# MetaGFN: Exploring Distant Modes with Adapted Metadynamics for Continuous GFlowNets

**Dominic Phillips**                                          *dominic.phillips@ed.ac.uk*
*University of Edinburgh, Edinburgh*

**Flaviu Cipcigan**                                          *flaviu@flaviucipcigan.com*
*IBM Research Europe, UK*

**Reviewed on OpenReview:** *https://openreview.net/forum?id=dtyNeemB7A*

## Abstract

Generative Flow Networks (GFlowNets) are a class of generative models that sample objects in proportion to a specified reward function through a learned policy. They can be trained either on-policy or off-policy, needing a balance between exploration and exploitation for fast convergence to a target distribution. While exploration strategies for discrete GFlowNets have been studied, exploration in the continuous case remains to be investigated, despite the potential for novel exploration algorithms due to the local connectedness of continuous domains. Here, we introduce Adapted Metadynamics, a variant of metadynamics that can be applied to arbitrary black-box reward functions on continuous domains. We use Adapted Metadynamics as an exploration strategy for continuous GFlowNets. We show several continuous domains where the resulting algorithm, MetaGFN, accelerates convergence to the target distribution and discovers more distant reward modes than previous off-policy exploration strategies used for training GFlowNets.

## 1 Introduction

Generative Flow Networks (GFlowNets) are a type of generative model that samples from a discrete space $\mathcal{X}$ by sequentially constructing objects via a sequence of actions taken from a learned policy $P_F$ (Bengio et al., 2021a). The policy $P_F(\mathbf{s}, \mathbf{s}')$ specifies the probability of transitioning from some intermediate state $\mathbf{s}$ to some other state $\mathbf{s}'$. The policy is parameterised and trained so that, at convergence, the probability of eventually sampling an object $\mathbf{x} \in \mathcal{X}$ is proportional to a specified reward function $R(\mathbf{x})$.

GFlowNets offer advantages over more traditional sampling methods, such as Markov chain Monte Carlo (MCMC), by learning an amortised sampler, capable of single-shot generation of samples from the desired distribution. Since GFlowNets learn a parametric policy, they can generalise across states, resulting in higher performance across various tasks (Bengio et al., 2021a; Malkin et al., 2022; Zhang et al., 2022; Jain et al., 2022; Deleu et al., 2022; Jain et al., 2023; Hu et al., 2023; Zhang et al., 2023; Shen et al., 2023b). Applications explored so far include conditioned molecular generation (Shen et al., 2023b), maximum likelihood estimation in discrete latent variable models (Hu et al., 2023), structure learning of Bayesian networks (Deleu et al., 2022), scheduling computational operations (Zhang et al., 2023), and discovering reticular materials for carbon capture (Cipcigan et al., 2023).

Although originally conceived for discrete state spaces, GFlowNets have been extended to more general state spaces, such as entirely continuous spaces, or spaces that are hybrid discrete-continuous (Lahlou et al., 2023). In the continuous setting, given the current state, the policy specifies a continuous probability distribution over subsequent states, and the probability density over states $\mathbf{x} \in \mathcal{X}$ sampled with the policy is proportional to a reward density function $r(\mathbf{x})$. The continuous domain opens up more applications for GFlowNets, such as molecular conformation sampling (Volokhova et al., 2023) and continuous control problems (Luo et al., 2024).

GFlowNets are trained similar to reinforcement learning agents. Trajectories of states are generated either on-policy or off-policy, with the terminating state $\mathbf{x} \in \mathcal{X}$ providing a reward signal to inform a gradient step on the policy parameters. GFlowNets therefore suffer from the same training pitfalls as reinforcement learning. One such issue is slow temporal credit assignment, which has thus far been addressed by designing more effective loss functions, such as detailed balance (Bengio et al., 2021b), trajectory balance (Malkin et al., 2022) and sub-trajectory balance (Madan et al., 2022). This latter approach has recently been extended by providing an energy function inductive bias at the early stages of training (Pan et al., 2023).

In addition to the choice of loss function, GFlowNet training also depends on the exploration strategy used to collect samples. Sampling purely on-policy learning is usually inefficient, as it can quickly become trapped in a local optimum and fail to discover novel, high-reward objects. More successful strategies, therefore, rely on a degree of off-policy exploration. For the discrete setting, numerous exploration strategies have been proposed, including $\epsilon$-noisy with a uniform random policy, tempering, Generative Augmented Flow Networks (GAFN) (Pan et al., 2022), Thompson sampling (Rector-Brooks et al., 2023), and Local Search GFlowNets (Kim et al., 2024). Although these approaches can be generalised to the continuous domain, there is limited literature benchmarking their effectiveness in this setting.

Sampling in the continuous setting is a common occurrence in various domains such as molecular modelling (Hawkins, 2017; Yang et al., 2019) and Bayesian inference (Shahriari et al., 2016). The local connectedness of a continuous domain allows for novel exploration strategies that are not directly applicable in the discrete setting. Sendera et al. (2024) compared several exploration strategies in the context of diffusion samplers and proposed alternating between on-policy sampling and backward sampling from a fixed set of MCMC samples when training continuous GFlowNets. These samples were selected prior to training, thereby inducing a fixed pre-training cost but keeping the cost of each training trajectory constant. The authors showed effective exploration, even in high-dimensional problems. However, their MCMC approach requires access to cheap gradients of the reward landscape, which may not always be readily available (Rengarajan et al., 2022). In addition, there are some settings, such as in molecular conformation sampling, where MCMC approaches are known to require significantly longer timescales to overcome energy barriers than methods based on molecular dynamics (MD) simulations (Abrams and Bussi, 2014), thereby making the pre-training cost prohibitively expensive.

We introduce MetaGFN, a new exploration algorithm for continuous GFlowNets inspired by metadynamics, an enhanced sampling method widely used in molecular modeling (Laio and Parrinello, 2002). Unlike MCMC-based approaches, MetaGFN operates in a general black-box setting: it queries only reward values from an oracle and does not require reward gradients. As in standard metadynamics, MetaGFN converges rapidly when a low-dimensional representation of the reward function, specified through a so-called *collective variable* (CV) basis, is known. This makes it particularly effective for small molecular systems, where such CVs are well established and typically low dimensional (Fiorin et al., 2013). Moreover, MetaGFN requires no pretraining and adds only a small, constant overhead per training trajectory, which becomes negligible when the cost of evaluating the policy or reward is non-trivial, which is the case if either of these is implemented via a deep learning model.

The main contributions of this work are:

- Introducing MetaGFN, an algorithm that adapts metadynamics to black box rewards and continuous GFlowNets;

- Proving that the Adapted Metadynamics formulation underlying MetaGFN is consistent and reduces to standard metadynamics in the appropriate limit;

- Showing empirically that MetaGFN outperforms existing GFlowNets exploration strategies in various continuous environments, including alanine dipeptide conformation sampling.

The rest of the paper is as follows. In Section 2 we review the theory of discrete and continuous GFlowNets, as well as metadynamics and collective variables. In Section 3, we present the Adapted Metadynamics and MetaGFN algorithms. In Section 4, we evaluate MetaGFN against other approaches, showing that MetaGFN generally outperforms existing exploration strategies in various continuous environments. We

finish with limitations and conclusions in Sections 5 and 6. Code for MetaGFN is available at `https://github.com/dominicp6/metad_contgfn.git`.

## 2 Preliminaries

### 2.1 Discrete GFlowNets

In a GFlowNet, the *network* refers to a directed acyclic graph (DAG), denoted $G = (\mathcal{S}, \mathcal{A})$. Nodes represent *states* $\mathbf{s} \in \mathcal{S}$, and edges represent *actions* $\mathbf{s} \to \mathbf{s}' \in \mathcal{A}$ denoting one-way transitions between states. The DAG has two distinguishable states: a unique *source state* $\mathbf{s}_0$, which has no incoming edges, and a unique *sink state* $\bot$, that has no outgoing edges.

The set of states $\mathcal{X} \subset \mathcal{S}$ that are directly connected to the sink state are known as *terminating states*. GFlowNets learn forward transition probabilities, known as a *forward policy* $P_F(\mathbf{s}'|\mathbf{s})$, along the edges of the DAG so that the resulting marginal distribution over the terminal states complete trajectories (the *terminal distribution*) , denoted as $P^\bot(\mathbf{x})$, is proportional to a given *reward function* $R : \mathcal{X} \to \mathbb{R}$. GFlowNets also introduce additional learnable objects, such as a *backward policy* $P_B(\mathbf{s}|\mathbf{s}')$, which is a distribution over the parents of any state of the DAG, to create losses that train the forward policy. Objective functions for GFlowNets include flow matching (FM), detailed balance (DB), trajectory balance (TB) and subtrajectory balance (STB) (Bengio et al., 2021a;b; Malkin et al., 2022; Madan et al., 2022). During training, the parameters of the flow objects are updated with stochastic gradients of the objective function applied to batches of trajectories. These trajectory batches can be obtained either directly from the current forward policy or from an alternative algorithm that encourages exploration. These approaches are known as *on-policy* and *off-policy* training respectively.

### 2.2 Continuous GFlowNets

Continuous GFlowNets extend the generative problem to continuous spaces (Lahlou et al., 2023), where the quantity analogous to the DAG is a measurable pointed graph (MPG) (Nummelin, 1984). MPGs can model continuous spaces (e.g., Euclidean space, spheres, tori), as well as hybrid spaces, with a mix of discrete and continuous components, as often encountered in robotics, finance, and biology (Bortolussi and Policriti, 2008; Swiler et al., 2012; Neunert et al., 2020).

**Definition 2.1** (Measurable pointed graph (MPG)). Let $(\bar{\mathcal{S}}, \mathcal{T})$ be a topological space, where $\bar{\mathcal{S}}$ is the *state space*, $\mathcal{T}$ is the set of open subsets of $\bar{\mathcal{S}}$, and $\Sigma$ is the Borel $\sigma$-algebra associated with the topology of $\bar{\mathcal{S}}$. Within this space, we identify the *source state* $\mathbf{s}_0 \in \bar{\mathcal{S}}$ and *sink state* $\bot \in \bar{\mathcal{S}}$, both distinct and isolated from the rest of the space. In this space, we define a *reference transition kernel* $\kappa : \bar{\mathcal{S}} \times \Sigma \to [0, +\infty)$ and a *backward reference transition kernel* $\kappa^b : \bar{\mathcal{S}} \times \Sigma \to [0, +\infty)$. The support of $\kappa(\mathbf{s}, \cdot)$ are all open sets accessible from $\mathbf{s}$. The support of $\kappa^b(\mathbf{s}, \cdot)$ are all open sets from which $\mathbf{s}$ is accessible. Additionally, these objects must be well-behaved in the following sense:

  (i) *Continuity*: For all $B \in \Sigma$, the mapping $\mathbf{s} \mapsto \kappa(\mathbf{s}, B)$ is continuous.

  (ii) *No way back from the source*: The backward reference kernel has zero support at the source state, that is, for all $B \in \Sigma$, $\kappa^b(\mathbf{s}_0, B) = 0$.

  (iii) *No way forward from the sink*: When at the sink, applying the forward kernel keeps you there, that is, $\kappa(\bot, \cdot) = \delta_\bot(\cdot)$, where $\delta_\bot$ is the Dirac measure of the sink state.

  (iv) *A fully-explorable space*: The number of steps required to possibly reach any measurable $B \in \Sigma$ from the source state, and to guarantee to reach the sink state, with the forward reference kernel is bounded.

The set of objects $(\bar{\mathcal{S}}, \mathcal{T}, \Sigma, \mathbf{s}_0, \bot, \kappa, \kappa^b)$ then defines an MPG.

Note that the support of $\kappa(\mathbf{s}, \cdot)$ and $\kappa^b(\mathbf{s}, \cdot)$ are analogous to the child and parent sets of a state $\mathbf{s}$ in a DAG.

The set of *terminating states* $\mathcal{X}$ are the states that can transition to the sink, given by $\mathcal{X} = \{\mathbf{s} \in \mathcal{S} : \kappa(\mathbf{s}, \{\perp\}) > 0\}$, where $\mathcal{S} := \bar{\mathcal{S}} \setminus \{\mathbf{s}_0\}$. *Trajectories* $\tau$ are sequences of states that run from source to sink, $\tau = (\mathbf{s}_0, \ldots, \mathbf{s}_n, \perp)$. The *forward Markov kernel* $P_F : \bar{\mathcal{S}} \times \Sigma \to [0, \infty)$ and *backward Markov kernel* $P_B : \bar{\mathcal{S}} \times \Sigma \to [0, \infty)$ have the same support as $\kappa(\mathbf{s}, \cdot)$ and $\kappa^b(\mathbf{s}, \cdot)$ respectively. As Markov kernels, $P_F(\mathbf{s}, \cdot)$ and $P_B(\mathbf{s}, \cdot)$ define probability distributions over next states and must therefore satisfy the normalisation condition: for all $\mathbf{s} \in \bar{\mathcal{S}}$, we have $\int_{\bar{\mathcal{S}}} P_F(\mathbf{s}, d\mathbf{s}') = \int_{\bar{\mathcal{S}}} P_B(\mathbf{s}, d\mathbf{s}') = 1$. A *flow* F is a tuple $F = (f, P_F)$, where $f : \Sigma \to [0, \infty)$ is a *flow measure*, satisfying $f(\{\perp\}) = f(\mathbf{s}_0) = Z$, where $Z$ is the *total flow*.

The *reward measure* is a positive and finite measure $R$ over the terminating states $\mathcal{X}$. The density of $R$ is denoted as $r(\mathbf{x})$, for $\mathbf{x} \in \mathcal{X}$. A flow $F$ is said to satisfy the *reward matching conditions* if

$$R(d\mathbf{x}) = f(d\mathbf{x}) P_F(\mathbf{x}, \{\perp\}).$$

If a flow satisfies the reward matching conditions and trajectories are generated by recursively sampling the Markov kernel $P_F$ starting at $s_0$, the resulting *measure over the terminating states*, $P^\perp(B)$, is proportional to the reward: $P^\perp(B) = \frac{R(B)}{R(\mathcal{X})}$ for any $B$ in the $\sigma$-algebra of terminating states (Lahlou et al., 2023).

Objective functions for discrete GFlowNets generalise to continuous GFlowNets. However, in the continuous case, the *forward policy* $\hat{p}_F : \mathcal{S} \times \bar{\mathcal{S}} \to [0, \infty)$, *backward policy* $\hat{p}_B : \mathcal{S} \times \bar{\mathcal{S}} \to [0, \infty)$ and *parameterised flow* $\hat{f} : \mathcal{S} \to [0, \infty)$ parameterise the $P_F$, $P_B$ transition kernels and flow measure $f$ on an MPG. Discrete GFlowNets parameterise log transition probabilities and flows on a DAG. In this work, we consider DB, TB and STB losses. For a complete trajectory $\tau$, the TB loss can be written as

$$L_{TB}(\tau) = \left( \log \frac{Z_{\boldsymbol{\theta}} \prod_{t=0}^{n} \hat{p}_F(\mathbf{s}_t, \mathbf{s}_{t+1}; \boldsymbol{\theta})}{r(\mathbf{s}_n) \prod_{t=0}^{n-1} \hat{p}_B(\mathbf{s}_{t+1}, \mathbf{s}_t; \boldsymbol{\theta})} \right)^2,$$

where $Z_{\boldsymbol{\theta}} \in \mathbb{R}$ is the parameterised total flow (see Appendix A for the DB and STB loss functions).

**Black-box continuous GFNs.** Later in this work, we introduce an exploration algorithm designed for training continuous GFNs in the *black-box setting*; where we do not have access to the internals of the function parameterisation of $r(x)$, hence a closed form expression for $\nabla r(x)$ is unavailable. Thus, if gradients of $r(x)$ are needed, they must typically be computed via finite differences, requiring $O(d)$ evaluations of $r(x)$, where $d = \dim(\mathcal{X})$. Computing $\nabla r(x)$ can then easily become a computational bottleneck. For instance, in drug discovery, $r(x)$ may quantify the binding affinity of a molecular conformer $x$ to a given target protein, and each evaluation of $r(x)$ might require costly molecular dynamics simulations of the protein system (e.g., via alchemical free-energy perturbation (Mey et al., 2020)).

## 2.3 Exploration strategies for GFlowNets

GFlowNets can reliably learn using off-policy trajectories, a key advantage over hierarchical variational models (Malkin et al., 2023). For optimal training, it is common to use a replay buffer and alternate between on-policy and off-policy (exploration) batches (Shen et al., 2023a). In the discrete case, proposed techniques to encourage exploration include $\epsilon$-noisy exploration, tempering, and the incorporation of intermediate rewards (Bengio et al., 2021a; Pan et al., 2022). Exploration strategies for continuous GFlowNets are less well studied in the literature, but several methods designed for discrete GFlowNets can be adapted (Sendera et al., 2024). In this work, we consider the following.

**Local Search GFlowNets** (Kim et al., 2024): Explores by backtracking and resampling on-policy trajectories. Reconstructed trajectories with a higher reward than the original are used for training, thus encouraging the learning of high-reward modes.

**Thompson sampling**. (Rector-Brooks et al., 2023): Explores high-uncertainty regions using an ensemble of policy heads with a shared torso. A random head generates the on-policy trajectory, and the loss is computed by averaging contributions over heads, where each head is independently included with probability $p$.

**Noisy exploration**: Explores by increasing policy uncertainty. A small constant is added to the policy variance which is gradually reduced to zero over the course of training.

**Nested sampling** (Lemos et al., 2023): A Markov-Chain Monte Carlo (MCMC) algorithm is used to sample from the reward distribution. A collection of these samples are stored in a replay buffer prior to training. Then, during training, off-policy trajectories are generated by backward sampling from these states using the current backward policy.

### 2.4 Metadynamics and collective Variables

Molecular dynamics (MD) simulates that dynamics of molecules using *Langevin dynamics* (LD) (Pavliotis, 2014), a stochastic differential equation that models the motion of particles under friction and random noise. LD trajectories ergodically sample the Gibbs measure of the molecule, $\rho_\beta(\mathbf{x}) \propto e^{-\beta V(\mathbf{x})}$, where $\mathbf{x} \in \mathcal{X}$ represents atomic positions, $V(\mathbf{x})$ is the molecular potential, and $\beta$ is the thermodynamic beta.[1] However, when $V(\mathbf{x})$ contains multiple, deep local minima, as is common in biomolecules, then LD becomes inefficient, as it tends to get trapped in these minima, slowing the exploration of the full state space.

*Metadynamics* overcomes this by progressively modifying the potential landscape to discourage visits to already explored regions (Laio and Parrinello, 2002)[2]. It achieves this by regularly depositing repulsive Gaussian biases, centered at the current state of the evolving LD trajectory. This modifies the potential to $V_{\text{total}}(\mathbf{x}, t) = V(\mathbf{x}) + V_{\text{bias}}(\mathbf{x}, t)$, where $V_{\text{bias}}(\mathbf{x}, t)$ is the cumulative bias at time $t$. Intuitively, metadynamics thus progressively transforms the landscape into a more level surface, in which the system can diffuse more freely, thereby accelerating exploration (Figure 1). In the limit $t \to \infty$, the dynamics approximates free diffusion and uniformly samples the domain of $V(\mathbf{x})$.

Since biases are typically specified on a numerical grid, this gives rise to a memory cost that grows exponentially with dimension. Therefore, biases are typically applied along low-dimensional coordinates known as *collective variables* (CVs). A collective variable $\mathbf{z}(\mathbf{x}) : \mathcal{X} \to \mathcal{Z}$ is any mapping from the original (high-dimension) state space $\mathcal{X}$ to a lower-dimensional space $\mathcal{Z}$. For potential $V(\mathbf{x})$ and collective variables $\mathbf{z}$, metadynamics is guaranteed to eventually uniformly sample the domain of the *marginal potential* $V(\mathbf{z}') := \int_{\mathcal{X}} \delta(\mathbf{z}' - \mathbf{z}(\mathbf{x})) V(\mathbf{x}) \mathrm{d}\mathbf{x}$. In molecular contexts, ideal CV choices correspond to the physical reaction coordinates that underly rare event transitions (e.g., backbone dihedrals, interatomic distances) (Laio and Gervasio, 2008; De Vivo et al., 2016). In more general settings, ideal CVs should resolve rare events dynamics on $V(x)$, simplfy the landscape (retaining its essential features, e.g. barriers and basins), and minimise the dimensionality of this representation. For simple systems, CVs can be derived from known symmetries or macroscopic order parameters that describe state changes. However, if the choice of CVs is not obvious, then they can learnt through data-driven methods such as Time-Lagged Independent Component Analysis (TICA) (Molgedey and Schuster, 1994), manifold learning algorithms, neural network autoencoders or variational methods (Mardt et al., 2018; Bonati et al., 2021; Ramaswamy et al., 2021). For an in-depth review on these CV-learning approaches, see Sidky et al. (2020). Metadynamics has seen numerous extensions (Bussi and Laio, 2020). In the next section, we adapt the original metadynamics algorithm to the continuous black-box setting. We will assume suitable CVs are given, but the theory we present is agnostic to their form–be it analytical expressions, neural networks, or non-parametric tabular mappings.

## 3 MetaGFN: Adapted Metadynamics for GFlowNets

Metadynamics has two properties that make it well-suited as an exploration strategy. Firstly, it eventually uniformly samples the domain, ensuring exploration of *all* local minima. In contrast, techniques such as Local Search GFlowNets, Thompson sampling, and noisy exploration rely on localised strategies that do not have global exploration guarantees and, as such, are prone to mode locking (Section 4). Secondly, since metadynamics gradually diffuses from the starting configuration, it allows for incremental exploration, which may result in more stable training than global approaches like nested sampling.

To apply metadynamics to GFlowNets, we must first translate the reward density into a potential energy function. We assume that $\mathcal{X}$ is a manifold and that the reward density $r(\mathbf{x})$ is bounded and L1 integrable

---

[1]We review Langevin dynamics in Appendix B.

[2]Note that the 'meta' in metadynamics stems from chemistry concepts, and is unrealed to meta-learning in computer science. The motivation for the name is that metadynamics accelerates transitions between minima (metastable states) of V(x).

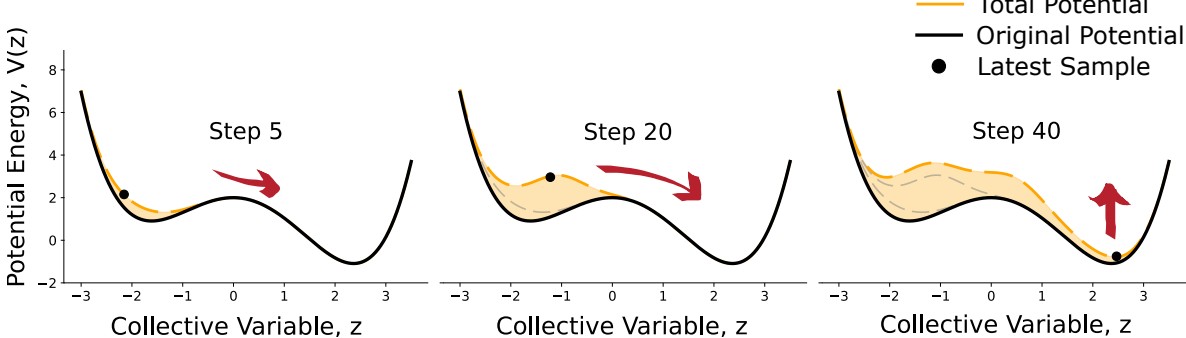

Figure 1: Illustration of metadynamics in a multi-well potential at three different snapshots in time. Regular deposition of a bias leads to a total potential that gradually flattens, encouraging exploration. Red arrows illustrate the general effect of the bias potential on the particle dynamics at that snapshot.

over $\mathcal{X}$ with at most finitely many discontinuities. Thus, the target density over the terminal states, $\rho(\mathbf{x}) := r(\mathbf{x})/\int_{\mathcal{X}} r(\mathbf{x}') \, d\mathbf{x}'$, can be expressed as a Gibbs distribution: $\rho(\mathbf{x}) = \exp(-\beta' V(\mathbf{x}))/\int_{\mathcal{X}} r(\mathbf{x}') \, d\mathbf{x}'$, where we identify $V(\mathbf{x}) = -\frac{1}{\beta'} \ln r(\mathbf{x})$, for some constant $\beta' > 0$. This formulation ensures that high-reward areas correspond to low-potential-energy minima. For simplicity, we set $\beta'$ equal to thermodynamic beta.[3]

However, similar to nested sampling, metadynamics has the drawback of requiring access to not only the potential, but also its gradients, which are not directly available in a black-box GFlowNet setting (Section 2.2). We now address this challenge by introducing two algorithms: *Adapted Metadynamics* (AM) for exploring the potential energy landscape to find high-reward states (Section 3.1) and *MetaGFN* for guiding the GFN to learn from these states and eventually sample all modes of the reward density (Section 3.2).

## 3.1 Adapted Metadynamics

Metadynamics requires the gradient of the total potential, where $-\nabla V_{\text{total}}(\mathbf{x}, t) = -\nabla(V(\mathbf{x}) + V_{\text{bias}}(\mathbf{x}, t))$. Using the above assumptions, we have $\nabla V(\mathbf{x}) = -\nabla r(\mathbf{x})/(\beta' r(\mathbf{x}))$. However, $r(\mathbf{x})$ is often a computationally expensive black-box function, and its gradient, $\nabla r(\mathbf{x})$, is unknown. While finite differences can estimate $\nabla r(\mathbf{x})$ for smooth, low-dimensional reward distributions, this approach is impractical in high-dimensional spaces. Below, we explain how to adapt metadynamics to this black-box setting and avoid finite difference gradient estimates by storing a dynamically-updated kernel density estimate (KDE) of the potential. We call the modified metadynamics algorithm *Adapted Metadynamics* (AM) (Algorithm 1).

Let $\mathbf{x}_t$ denote the metadynamics sample at time $t$ and $\mathbf{z}(\mathbf{x}) = (z_1(\mathbf{x}), \ldots, z_d(\mathbf{x}))$ be a given set of collective variables, where each $z_i$ is a one-dimensional coordinate, and $\mathbf{z} \in \mathcal{Z}$ is $d$-dimensional. Let $z_{i,t} := z_i(x_t)$ denote the corresponding $i^{th}$ CV coordinate at time $t$ and assume that the Jacobian $\nabla_{\mathbf{x}} \mathbf{z}$ is well defined for all $\mathbf{x}$. We store a discretisation of the KDE potential and bias potentials in CV space $\mathcal{Z}$. Since these are stored in memory, gradient computations are cheap and require no further evaluations of $r(\mathbf{x})$.

Let $\hat{V}(\mathbf{z}, t)$ represent the KDE of the marginal potential at time $t$. To compute $\hat{V}(\mathbf{z}, t)$, we maintain two separate KDEs: $\hat{N}(\mathbf{z}, t)$ for visited states (line 8) and $\hat{R}(\mathbf{z}, t)$ for cumulative rewards (line 9). We update these KDEs on-the-fly at the same time the bias potential is updated, which occurs every integer $n$ steps of Langevin dynamics (line 7). If $\mathcal{Z}$ is isomorphic to Euclidean space, we use Gaussian kernels with kernel width $\boldsymbol{\sigma} = (\sigma_1, \ldots, \sigma_k) \in \mathbb{R}^k$, matching the width of the Gaussian bias (line 11)[4]. This is a reasonable as both are set by the variability length scale of $V(\mathbf{x})$. Finally, the KDE potential $\hat{V}(\mathbf{z}, t)$ is then calculated as $\hat{V}(\mathbf{z}, t) = -\frac{1}{\beta} \log \left( \frac{\hat{R}(\mathbf{z}, t)}{\hat{N}(\mathbf{z}, t) + \epsilon} + \epsilon \right)$, for a fixed constant $\epsilon > 0$ (line 10). We found empirically that $\epsilon$ ensured numerical stability by preventing division by zero and bounding $\hat{V}$ from above. Defining $\hat{V}$ through the ratio

---

[3]Making this choice means that the single parameter $\beta$ uniquely controls the (unbiased) transition rates between minima of the potential - from the Kramer formula - transition rate $\propto \frac{1}{\beta} \exp(\beta \Delta V)$, where $\Delta V \propto \frac{1}{\beta'} = \frac{1}{\beta}$.

[4]If $\mathcal{Z}$ is isomorphic to a $d$-torus, we use von Mises distributions.

---

**Algorithm 1:** Adapted Metadynamics

---

**Input** : Manifold environment of terminating states $\mathcal{X}$ with reward density $r : \mathcal{X} \to \mathbb{R}$. Initial state $(\mathbf{x}_t, \mathbf{p}_t) \in \mathcal{X} \times T_{\mathbf{x}_t}(\mathcal{X})$. Collective variables $\mathbf{z} = (z_1, \ldots, z_d)$.

**Parameters** : Gaussian width $\boldsymbol{\sigma} = (\sigma_1, \ldots, \sigma_d) \in \mathbb{R}^d$. Gaussian height $w > 0$. Stride $n \in \mathbb{Z}^+$. LD parameters: $\gamma, \beta$. Timestep $\Delta t$.

**1** $\hat{N} \leftarrow 0$

**2** $\hat{R} \leftarrow 0$

**3** $\hat{V}(\mathbf{z}) \leftarrow 0$

**4** $V_{\text{bias}}(\mathbf{z}) \leftarrow 0$

**5** **every** timestep $\Delta t$:

**6**     $\mathbf{z}_t \leftarrow \mathbf{z}(\mathbf{x}_t)$

**7**     **every** n timesteps $n\Delta t$:

**8**         $\hat{N} \leftarrow \hat{N} + \exp\left(-\frac{1}{2}\sum_{i=1}^d \frac{(z_i - z_{i,t})^2}{\sigma_i^2}\right)$

**9**         $\hat{R} \leftarrow \hat{R} + r(\mathbf{x}_t) \cdot \exp\left(-\frac{1}{2}\sum_{i=1}^d \frac{(z_i - z_{i,t})^2}{\sigma_i^2}\right)$

**10**         $\hat{V} \leftarrow -\frac{1}{\beta} \log\left(\hat{R}/(\hat{N} + \epsilon) + \epsilon\right)$

**11**         $V_{\text{bias}}(\mathbf{z}) \leftarrow V_{\text{bias}}(\mathbf{z}) + n \cdot \Delta t \cdot w \cdot \exp\left(-\frac{1}{2}\sum_{i=1}^d \frac{(z_i - z_{i,t})^2}{\sigma_i^2}\right)$

**12**     **compute** forces:

**13**         $F \leftarrow -\left(\nabla_{\mathbf{z}}\hat{V}(\mathbf{z})|_{\mathbf{z}=\mathbf{z}_t} + \nabla_{\mathbf{z}}V_{\text{bias}}(\mathbf{z})|_{\mathbf{z}=\mathbf{z}_t}\right) \cdot \nabla_{\mathbf{x}}\mathbf{z}|_{\mathbf{x}=\mathbf{x}_t}$

**14**     **propagate** $\mathbf{x}_t, \mathbf{p}_t$ by $\Delta t$ using Langevin dynamics with computed force $F$ (Alg. 3, Appendix B).

---

$\hat{R}/\hat{N}$ ensures that it rapidly and smoothly adjusts whenever new modes are discovered. Furthermore, we prove that $\hat{V}$ eventually discovers *all* reward modes in the CV space. More precisely,

**Theorem 3.1.** *If the collective variable $\boldsymbol{z}(\boldsymbol{x})$ is analytic with a bounded domain, then*

$$\lim_{\epsilon \to 0} \lim_{\boldsymbol{\sigma} \to \mathbf{0}} \lim_{t \to \infty} \hat{V}(\boldsymbol{z}, t) = V, \tag{1}$$

*where $V = V(\boldsymbol{z}') := \int_{\mathcal{X}} \delta(\boldsymbol{z}' - \boldsymbol{z}(\boldsymbol{x})) V(\boldsymbol{x}) \mathrm{d}\boldsymbol{x}$ is the ground truth free-energy surface in CV space.*

The proof is in Appendix C.

Note that the algorithm can be extended to a batch of trajectories, where each metadynamics trajectory evolves independently, but with a shared $\hat{V}$ and $V_{\text{bias}}$ which receive updates from every trajectory in the batch. We found empirically that this accelerates exploration and reduces stochastic gradient noise during training and this is the version we use in our experiments (Section 4).

## 3.2 MetaGFN

Each Adapted Metadynamics sample $\mathbf{x}_i \in \mathcal{X}$ is an off-policy terminal state sample. To train a GFlowNet, complete trajectories are required. We generate these by backward sampling from the terminal state, giving a trajectory $\tau = (\mathbf{s}_0, \mathbf{s}_1, \ldots, \mathbf{s}_n = \mathbf{x}_i)$, where each state $\mathbf{s}_{i-1}$ is sampled from the current backward policy distribution $\hat{p}_B(\mathbf{s}_{i-1}|\mathbf{s}_i; \boldsymbol{\theta})$, for $i$ from $n$ to 1. This approach means that the generated trajectory $\tau$ has reasonable credit according to the loss function, thus providing a useful learning signal. However, since this requires a backward policy, this is compatible with DB, TB, and STB losses, but not FM loss. Given the superior credit assignment of the former losses, this is not a limitation (Madan et al., 2022).

In addition, we use a replay buffer. Due to the theoretical guarantee that AM will eventually sample all collective variable space (Theorem 3.1), AM samples are ideal candidates for storing in a replay buffer. When storing these trajectories in the replay buffer, there are two obvious choices: (1) Store the entire trajectory the first time it is generated, (2) Store only the AM sample and regenerate trajectories using the current backward policy when retrieving from the replay buffer. We investigated both options in preliminary experiments

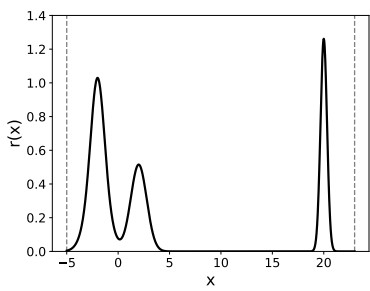

Figure 2: Line environment reward density.

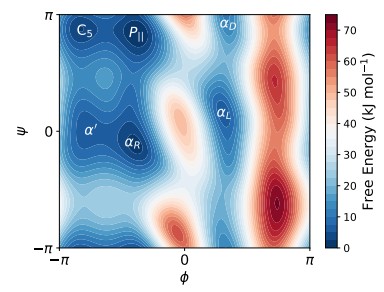

Figure 3: Free energy surface of alanine dipeptide.

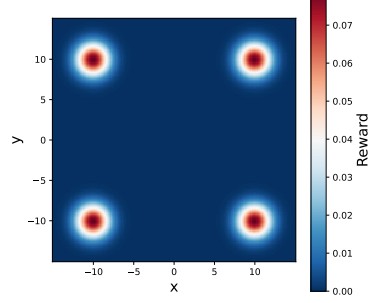

Figure 4: Grid environment reward density in dimension 2.

(Appendix D.2). Option 2 was found to be uniformly superior and it is the version we use in our main experiments (Section 4). We call the overall training algorithm *MetaGFN*, with pseudocode presented in Algorithm 2, below.

---

**Algorithm 2:** MetaGFN

**Input**   : Forward policy $P_F$. Backwards policy $P_B$. Loss function $L$.

**Parameters:** How often to run Adapted Metadynamics batches, `freqMD`. How often to run replay buffer batches, `freqRB`. Batch size, $b$. Stride, $n \in \mathbb{Z}^+$. Time step, $\Delta t > 0$

1 **for** each episode **do**:
2  **if** episode number is divisible by `freqMD`:
3   Run Adapted Metadynamics (batch size $b$) for time $n\Delta t$, obtain samples $\{\mathbf{x}_1, \ldots, \mathbf{x}_b\}$
4   Push $\{\mathbf{x}_1, \ldots, \mathbf{x}_b\}$ to the replay buffer
5   Backward sample from $\{\mathbf{x}_1, \ldots, \mathbf{x}_b\}$ using current $P_B$ to obtain trajectories $\{\tau_1, \ldots, \tau_b\}$
6  **elif** episode number is divisible by `freqRB`:
7   Random sample $\{\mathbf{x}_1, \ldots, \mathbf{x}_b\}$ from the replay buffer
8   Backward sample from $\{\mathbf{x}_1, \ldots, \mathbf{x}_b\}$ using current $P_B$ to obtain trajectories $\{\tau_1, \ldots, \tau_b\}$
9  **else**:
10   Generate trajectories $\{\tau_1, \ldots, \tau_b\}$ on-policy
11  Compute loss $l = \sum_{i=1}^{b} L(\tau_i, P_F, P_B)$
12  Take gradient step on loss $l$

---

## 4 Experiments

We compare MetaGFN with Thompson sampling, noisy, Local Search GFlowNets and nested sampling (Section 2.3). We run experiments in five continuous environments, summarised below. For each exploration strategy, we use a replay buffer and alternate between exploration and replay buffer batches. For MetaGFN, we use `freqRB = 2`, `freqMD = 10`. The forward and backward kernels are Gaussian/von Mises mixture distributions, with distribution parameters specified at each state by an MLP. In all environments, the additional computational expense of running Adapted Metadynamics was negligible ($<5\%$) compared to the training time of the models. Full experimental details are in Appendix D.

**Line environment**: A one-dimensional environment with state space $\mathcal{S} = \mathbb{R} \times \{t \in \mathbb{N}, 1 \leq t \leq 3\}$, where $t$ indexes the position of a state in a trajectory. The source state is $s_0 = (0, 0)$. The terminal states are therefore $\mathcal{X} = \mathbb{R} \times \{3\} \cong \mathbb{R}$. The collective variable is the identity, i.e. $z = x$. The reward density, plotted in Figure 2, consists of an asymmetric bimodal peak near the origin and an additional distant lone peak. It is

given by the Gaussian mixture distribution:

$$r(x) = \begin{cases} \mathcal{N}(-2.0, 1.0) + \mathcal{N}(-2.0, 0.4) + \\ \quad \mathcal{N}(2.0, 0.6) + \mathcal{N}(20.0, 0.1); & -5 \leq x \leq 23 \\ 0; & \text{otherwise,} \end{cases} \tag{2}$$

where $\mathcal{N}(\mu, \sigma^2)$ is a Gaussian density with mean $\mu$ and variance $\sigma^2$.

**Alanine dipeptide environment**: One application of continuous GFlowNets is molecular conformation sampling (Volokhova et al., 2023). Here, we train a GFlowNet to sample conformational states of alanine dipeptide (AD), a small biomolecule of 23 atoms that plays a key role in modelling the dynamics of proteins (Hermans, 2011). The metastable states of AD are distinguished in a 2D CV space defined by the backbone dihedral angles $\phi$ and $\psi$. The resulting free energy surface $V(\phi, \psi)$ in explicit water, obtained after extensive sampling with long molecular dynamics simulations, is shown in Figure 3. The metastable states, in increasing energy, are $P_{||}$, $\alpha_R$, $C_5$, $\alpha'$, $\alpha_L$, and $\alpha_D$. The state space is $\mathcal{S} = \mathbb{T}^2 \times \{t \in \mathbb{N}, 1 \leq t \leq 3\}$, with source state $s_0 = P_{||} = (-1.2, 2.68)$. Terminal states are $\mathcal{X} = \mathbb{T}^2 \times \{3\} \cong \mathbb{T}^2$. The reward function is the Boltzmann weight, $r(\phi, \psi) = \frac{1}{Z} \exp(-\beta V(\phi, \psi))$, where $Z$ is the normalisation constant.

**Grid environments**: We consider (hyper)grids in $d = 2$, 3 and 4 dimensions. The state space is $\mathcal{S} = [-15, 15]^d \times \{t \in \mathbb{N}, 1 \leq t \leq 3\}$. The $k$-dimensional (hyper)grid consists of $2^k$ modes, located at the corners of the (hyper)cube $[-10, 10]^d$. The (hyper)grid is centered on the origin with an edge width of 20. The reward modes are Gaussians with variance $\sigma^2 = 2$ (Figure 4). We use trajectory lengths of 3, 5, and 6 for dimensions 2, 3, and 4 respectively. The collective variable is the identity, i.e. $\mathbf{z} = \mathbf{x}$. The source state is the origin and terminal states are in $\mathcal{X} \cong \mathbb{R}^k$.

### 4.1 Results

In each environment, we run experiments with three different loss functions: Detailed Balance (DB), Trajectory Balance (TB) and Subtrajectory Balance (STB). We evaluated performance by computing the L1 error between the known reward distribution and the empirical on-policy distribution during training resulting from $10^4$ independent samples. The results, averaged over 10 repeats, are shown in Figure 5. In 12 of 15 experiments, MetaGFN outperformed all other exploration strategies. We give further analysis below.

*Line Environment*: Among losses, TB shows the lowest variance and, for all losses, MetaGFN always converges to the lowest error[5]. Indeed, MetaGFN is the only method that consistently samples the distant reward peak at $x = 20$ (Appendix D.2). Although other strategies occasionally sampled the distant peak, MetaGFN alone converges because it *continuously* samples this peak, even if the forward policy starts to lock onto the central modes, thus ensuring that the replay buffer is always populated with diverse samples. The slight increase in the loss of MetaGFN around batch number $5 \times 10^3$ occurs as the on-policy distribution widens when Adapted Metadynamics first discovers the distant peak. Appendix D.2 provides further analysis of Adapted Metadynamics in this environment and compares different MetaGFN variants, with and without noise, and with and without trajectory regeneration. We confirm that the variant of MetaGFN presented in Figure 5 (no added noise, always regenerate trajectories) is the most robust.

*Alanine Dipeptide Environment*: For each loss, MetaGFN generally converges to a lower error than all other exploration strategies. However, for TB loss, the average L1 error is marginally higher than on-policy training, but this conceals the fact that the best-case error is smaller. To better understand this result, we examined the best and worst training runs (as measured by L1 error) for TB on-policy and TB MetaGFN, shown in Figure 6. Note that, unlike on-policy, the best MetaGFN run learns to sample the rare $\alpha_L$ mode. In the worst case, MetaGFN fails to converge (although this is rare; only one of 10 runs failed). In Table 1, we quantify how often the different AD modes were sampled over the different repeats (a mode is considered sampled if the on-policy distribution has a mode within the correct basin of attraction). TB loss with MetaGFN is the only combination that consistently samples the majority of modes. The only mode not sampled by any method is $\alpha_D$, which has a natural abundance approximately 10 times less frequent than $\alpha_L$.

---

[5]We do not show results for nested sampling on the line environment as the implementation used did not support one-dimensional environments, see Appendix D.

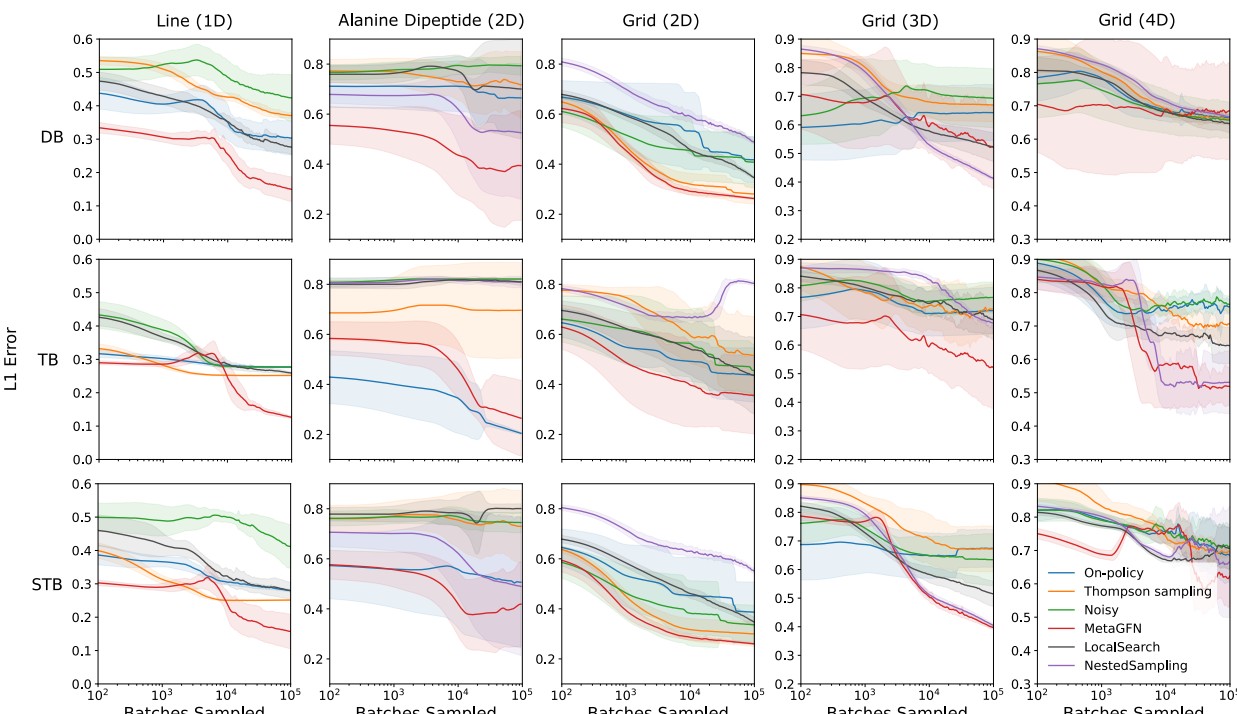

Figure 5: The L1 difference between on-policy and reward distribution during training for different loss functions and exploration strategies. The mean is plotted with standard error over 10 repeats. DB - Detailed Balance loss, TB - Trajectory Balance loss, STB - Subtrajectory Balance loss.

Table 1: Number of correct samples of AD modes in trained GFlowNets over 10 independent repeats for DB, STB, and TB loss functions. OP - On-policy and MD - MetaGFN. The $\alpha_D$ mode wasn't sampled in any model due to its low natural frequency.

|  | DB | | STB | | TB | |
|---|---|---|---|---|---|---|
|  | OP | MD | OP | MD | OP | MD |
| $P_{\parallel}$ | 1 | **7** | **6** | 5 | 10 | **8** |
| $\alpha_R$ | 6 | **9** | 7 | **10** | 10 | **9** |
| $C_5$ | 2 | **7** | 5 | **6** | 10 | **8** |
| $\alpha'$ | 6 | **9** | 5 | **10** | 5 | **9** |
| $\alpha_L$ | 0 | **1** | **1** | 0 | 0 | **8** |

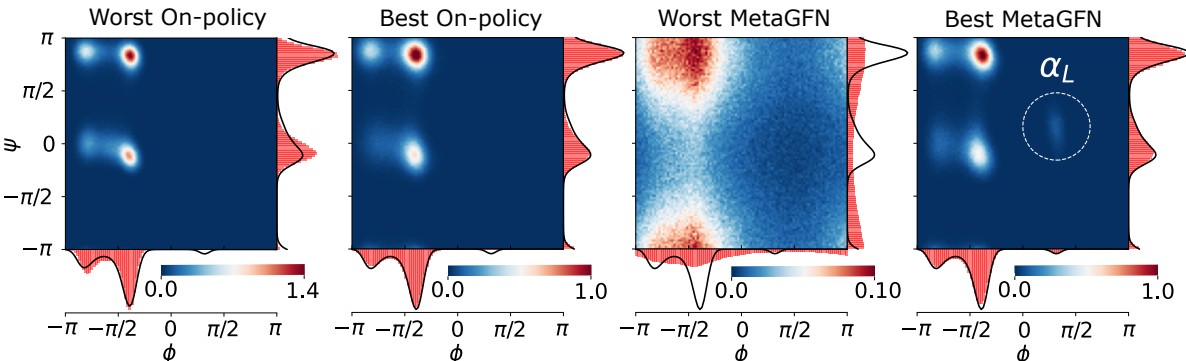

Figure 6: Learned on-policy distribution for TB on-policy and TB MetaGFN training runs. The colour bar shows the probability density. Red histograms show the marginal distribution along the angular coordinates. Black curves show the marginal distributions of the ground truth. In the best case, MetaGFN is able to learn the $\alpha_L$ mode. In the worst case, MetaGFN fails to converge. On-policy training, although more consistent, fails to learn to sample from the $\alpha_L$ mode.

*Grid Environments*: MetaGFN achieves the lowest error in 7 of 9 experiments, although note that the performance of all methods generally decreases with increasing dimension. This is unsurprising, since the exploration task becomes more difficult. For MetaGFN, this is also an expected outcome of the curse of dimensionality of the replay buffer grid; in high-dimensional spaces metadynamics samples encounter high-reward modes less frequently, leading to a replay buffer with less sample diversity. This is a known limitation of metadynamics (Laio and Parrinello, 2002). Nested Sampling outperforms MetaGFN in one case (Grid 3D, DB loss) but populating the replay buffer with nested samples from the terminal distribution relies on costly MCMC-based pretraining. The cost of this pretraining scales exponentially with dimension and requires reward gradients, unlike MetaGFN. The generally favourable performance of MetaGFN could be due to the method discovering an overall larger diversity of off-policy samples, that are refreshed and updated during training, in contrast to the static samples when using Nested Sampling approach.

## 5   Limitations

For metadynamics to be an effective sampler, the CVs must be low-dimensional and have bounded support. If they are not low-dimensional, then the memory cost of storing the CVs on the bias grid is too prohibitive, growing exponentially with dimension. If they have unbounded support, then exploration is not guaranteed to converge. Additionally, either these CVs should be known analytically (as in our experiments) or learnt from data (Sidky et al., 2020). Alternatively, CVs could be adaptively learnt by parameterising $\mathbf{z}(\mathbf{x}; \boldsymbol{\theta})$ by a neural network and updating its parameters by back-propagating through the GFlowNet loss during training. The metadynamics algorithm could also be replaced with a variant with smoother convergence properties or slightly better dimension scaling, such as well-tempered metadynamics (Barducci et al., 2008), reconnaissance metadynamics (Tribello et al., 2010), or on-the-fly probability-enhanced sampling (OPES) (Invernizzi, 2021). The bias potential could also be defined parametrically, lowering memory overhead. We leave these as extensions for future work.

## 6   Conclusions

While exploration strategies for discrete Generative Flow Networks (GFlowNets) have been widely studied, methodologies for continuous GFlowNets remain largely unexplored. To bridge this gap, we demonstrated how metadynamics, a well-established enhanced sampling technique in molecular dynamics, can be adapted as an effective exploration strategy for continuous GFlowNets.

In molecular dynamics, atomic forces are derived from the gradient of the potential, whereas continuous GFlowNets address problems where the reward function is a black box, making gradients generally inaccessible.

We showed that this challenge can be overcome by dynamically updating a kernel density estimate of the reward function, proving that this approach ensures proper exploration in the appropriate limit. Our empirical results highlight that MetaGFN provides a computationally efficient way to discover new modes in environments where prior knowledge of collective variables is available.

More broadly, this work advocates for leveraging techniques from molecular modeling to advance machine learning. Looking ahead, we anticipate that this cross-disciplinary perspective will open new directions, enabling further applications of enhanced sampling methods in generative modeling and reinforcement learning.

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

## A  Loss functions

For a complete trajectory $\tau$, the *detailed balanced loss* (DB) is

$$L_{DB}(\tau) = \sum_{t=0}^{n-1} \left( \log \frac{\hat{f}(\mathbf{s}_t; \boldsymbol{\theta}) \hat{p}_F(\mathbf{s}_t, \mathbf{s}_{t+1}; \boldsymbol{\theta})}{\hat{f}(\mathbf{s}_{t+1}; \boldsymbol{\theta}) \hat{p}_B(\mathbf{s}_{t+1}, \mathbf{s}_t; \boldsymbol{\theta})} \right)^2,$$

where $\hat{f}(\boldsymbol{s}_{t+1}; \boldsymbol{\theta})$ is replaced with $r(\mathbf{s}_n)$ if $\mathbf{s}_n$ is terminal.

The *subtrajectory balance loss* (STB) is

$$L_{STB}(\tau) = \frac{\sum_{0 \le i < j \le n} \lambda^{j-1} \mathcal{L}_{TB}(\tau_{i:j})}{\sum_{0 \le i < j \le n} \lambda^{j-i}},$$

$$\mathcal{L}_{STB}(\tau_{i:j}) := \left( \log \frac{\hat{f}(\mathbf{s}_i; \boldsymbol{\theta}) \prod_{t=i}^{j-1} \hat{p}_F(\mathbf{s}_{t+1}|\mathbf{s}_t; \boldsymbol{\theta})}{\hat{f}(s_j; \boldsymbol{\theta}) \prod_{t=i+1}^{j} \hat{p}_B(\mathbf{s}_{t-1}|\mathbf{s}_t; \boldsymbol{\theta})} \right)^2,$$

where $\hat{f}(\mathbf{s}_j; \boldsymbol{\theta})$ is replaced with $r(\mathbf{s}_j)$ if $\mathbf{s}_j$ is terminal. In the above, $\lambda < 0$ is a hyperparameter. The limit $\lambda \to 0^+$ leads to average detailed balance. The $\lambda \to \infty$ limit gives the trajectory balance objective. We use $\lambda = 0.9$ in our experiments.

## B  Langevin dynamics

Langevin dynamics (LD), is defined through the Stochastic Differential Equation (SDE):

$$\mathrm{d}\mathbf{x} = \mathbf{M}^{-1}\mathbf{p}\mathrm{d}t \tag{3}$$

$$\mathrm{d}\mathbf{p} = \mathbf{f}(\mathbf{x})\mathrm{d}t - \gamma\mathbf{p}\mathrm{d}t + \sqrt{2\gamma\beta^{-1}}\mathbf{M}^{1/2}\mathrm{d}\mathbf{w}. \tag{4}$$

In the above, $\mathbf{x}, \mathbf{p} \in \mathbb{R}^D$ are vectors of instantaneous position and momenta respectively, $\mathbf{f} \colon \mathbb{R}^D \to \mathbb{R}^D$ is a force function, $\mathbf{w}(t)$ is a vector of $D$ independent Wiener processes, $\mathbf{M} \in \mathbb{R}^D \times \mathbb{R}^D$ is a constant diagonal mass matrix, and $\gamma, \beta > 0$ are constant scalars which can be interpreted as a friction coefficient and inverse temperature respectively. In conventional Langevin dynamics, the force function is given by the gradient of the potential energy function, $\mathbf{f}(\mathbf{x}) = -\nabla V(\mathbf{x})$, where $V \colon \mathbb{R}^D \to \mathbb{R}$ and the dynamics are ergodic with respect to the Gibbs-Boltzmann density

$$\rho_\beta(\mathbf{x}, \mathbf{p}) \propto e^{-\beta H(\mathbf{x}, \mathbf{p})},$$

where $H(\mathbf{x}, \mathbf{p}) = \mathbf{p}^T \mathbf{M}^{-1} \mathbf{p}/2 + V(\mathbf{x})$ is the Hamiltonian. Since the Hamiltonian is separable in position and momenta terms, the marginal Gibbs-Boltzmann density is position space is simply $\rho_\beta(\mathbf{x}) \propto e^{-\beta V(\mathbf{x})}$, often referred to as the *Gibbs density*.

We present the Euler-Maruyama numerican scheme for integrating equations 3 below. This is called in line 14 of Adapted Metadynamics (Algorithm 1).

---

**Algorithm 3:** Euler-Maruyama Langevin Dynamics Step

---

**Input** : Current state $(\mathbf{x}_t, \mathbf{p}_t)$. Force $\mathbf{f}$.

**Parameters**: Friction coefficient $\gamma$. Thermodynamic beta $\beta$. Timestep $\Delta t$.

**Output** : State $(\mathbf{x}_{t+\Delta t}, \mathbf{p}_{t+\Delta t})$ at the next timestep.

**1** Sample a random vector $R$, with the same dimension as $\mathbf{x}_t$, where each element is an independent sample from a standard normal.

**2** $\mathbf{x}_{t+\Delta t} = \mathbf{x}_t + \mathbf{p}_t \Delta t$

**3** $\mathbf{p}_{t+\Delta t} = \mathbf{p}_t + \mathbf{f}\Delta t - \gamma \mathbf{p}_t \Delta t + \sqrt{2\gamma \Delta t/\beta} \cdot R$

**4 return** $(\mathbf{x}_{t+\Delta t}, \mathbf{p}_{t+\Delta t})$

---

## C Proofs

**Lemma C.1.** *Let $(f_n(x))$ and $(g_n(x))$ be sequences of real functions where $\lim_{n\to\infty} f_n(x) = \infty$, $\lim_{n\to\infty} g_n(x) = \infty$ and $\lim_{n\to\infty} \frac{f_n(x)}{g_n(x)} = h(x)$. Then, for all $\epsilon > 0$, we have $\lim_{n\to\infty} \frac{f_n(x)}{g_n(x)+\epsilon} = h(x)$.*

*Proof.*

$$\lim_{n\to\infty} \frac{f_n(x)}{g_n(x) + \epsilon} = \lim_{n\to\infty} \frac{f_n(x)}{g_n(x)} \frac{1}{1 + \epsilon/g_n(x)}$$

and the right-hand side is the product of two functions whose limit exists so, by the product rule of limits

$$\lim_{n\to\infty} \frac{f_n(x)}{g_n(x)} \lim_{n\to\infty} \frac{1}{1 + \epsilon/g_n(x)} = h(x) \cdot 1 = h(x),$$

so done. $\qquad\square$

**Lemma C.2.** *Let $(X_i)$ be a sequence of continuous random variables that take values on a bounded domain $D \subset \mathbb{R}^d$ that asymptotically approaches the uniform random variable $U$ on $D$, i.e. $X_i \to U$ uniformly. Further, suppose*

$$h(x) := \frac{\sum_{i=1}^{\infty} f(x_i)g(x, x_i)}{\sum_{i=1}^{\infty} g(x, x_i)}$$

*exists, where $x_i \in D$ is a sample from $X_i$ and $f(x)$ and $g(x, x')$ are analytic functions on $D$ and $D \times D$ respectively. Then,*

$$h(x) = \frac{\sum_{i=1}^{\infty} f(u_i)g(x, u_i)}{\sum_{i=1}^{\infty} g(x, u_i)},$$

*where the $u_i$ are samples from $U$. We make no assumption of independence of samples.*

*Proof.* Fix a probability space $(\Omega, \mathcal{F}, P)$ on which $(X_i)$ and $U$ are defined. Recall that a continuous random variable $X$ that takes values on $D \subset \mathbb{R}^d$ is a measurable function $X : \Omega \to D$ where $(D, \mathcal{B})$ is a measure space and $\mathcal{B}$ is the Borel $\sigma$-algebra on $D$. Let $\omega \subset \Omega$ denote an arbitrary element of the sample space. The requirement that $X_i \to U$ uniformly can be written formally as:

$$\forall \epsilon > 0, \exists N(\epsilon) \text{ s.t. } \forall i > N(\epsilon), \forall \omega \subset \Omega, |X_i(\omega) - U(\omega)| < \epsilon.$$

We prove the Lemma by showing that equality holds for all possible sequences of outcomes $\omega_1, \omega_2, \ldots$. That is, we prove:

$$\frac{\sum_{i=1}^{\infty} f(X_i(\omega_i))g(x, X_i(\omega_i))}{\sum_{i=1}^{\infty} g(x, X_i(\omega_i))} = \frac{\sum_{i=1}^{\infty} f(U(\omega_i))g(x, U(\omega_i))}{\sum_{i=1}^{\infty} g(x, U(\omega_i))}. \tag{5}$$

Since these are ratios of infinite series, to prove their equality it is sufficient to show that the numerator of the LHS is asymptotically equivalent to the numerator of the RHS and that the denominator of the LHS is asymptotically equivalent to the denominator of the RHS. Recall that two sequences of real functions $(a_n)$

and $(b_n)$ are asymptotically equivalent if $\lim_{n\to\infty} \frac{a_n(x)}{b_n(x)} = c$ where $c$ is a constant. First, we prove that this holds with

$$a_n := \sum_{i=1}^{n} f(X_i(\omega_i))g(x, X_i(\omega_i)) \tag{6}$$

and

$$b_n := \sum_{i=1}^{n} f(U(\omega_i))g(x, U(\omega_i))). \tag{7}$$

We write

$$\frac{a_n}{b_n} = \frac{\sum_{i=1}^{N(\epsilon)} f(X_i(\omega_i))g(x, X_i(\omega_i)) + \sum_{i=N(\epsilon)+1}^{n} f(X_i(\omega_i))g(x, X_i(\omega_i))}{\sum_{i=1}^{N(\epsilon)} f(U(\omega_i))g(x, U(\omega_i))) + \sum_{i=N(\epsilon)+1}^{n} f(U(\omega_i))g(x, U(\omega_i)))}.$$

Dividing by $\sum_{i=N(\epsilon)+1}^{n} f(U(\omega_i))g(x, U(\omega_i)))$ and taking the limit $n \to \infty$ we have

$$\lim_{n\to\infty} \frac{a_n}{b_n} = \lim_{n\to\infty} \frac{\sum_{i=N(\epsilon)+1}^{n} f(X_i(\omega_i))g(x, X_i(\omega_i))}{\sum_{i=N(\epsilon)+1}^{n} f(U(\omega_i))g(x, U(\omega_i)))}.$$

Since $f$ and $g$ are analytic and $i > N(\epsilon)$ for all terms in the sums we have, by Taylor expansion, $f(X_i(\omega_i)) = f(U(\omega_i)) + O(\epsilon)$ and $g(x, X_i(\omega_i)) = g(x, U(\omega_i)) + O(\epsilon)$, hence

$$\lim_{n\to\infty} \frac{a_n}{b_n} = \lim_{n\to\infty} \left( 1 + \frac{nO(\epsilon)}{\sum_{i=N(\epsilon)+1}^{n} f(U(\omega_i))g(x, U(\omega_i)))} \right) = 1 + \lim_{n\to\infty} \frac{nO(\epsilon)}{O(n)} = 1 + O(\epsilon).$$

Finally, since $\epsilon$ can be made arbitrarily small by partitioning the sum at a $N(\epsilon)$ that is sufficiently large, we conclude that $\lim_{n\to\infty} \frac{a_n}{b_n} = 1$, hence $(a_n)$ and $(b_n)$ as defined in equation 6 and equation 7 are asymptotically equivalent. By a similar argument, it can be shown that

$$c_n := \sum_{i=1}^{n} g(x, X_i(\omega_i))$$

and

$$d_n := \sum_{i=1}^{n} g(x, U(\omega_i)))$$

are also asymptotically equivalent. This proves equation 5.

$\square$

Below, we present the proof of Theorem 3.1 that appears in the main text.

*Proof. For concreteness, throughout this proof we assume that the kernel function is a Gaussian. We explain at the appropriate stage in the proof, indicated by (\*), how this assumption can be relaxed.*

First we take the $n \to \infty$ limit. Recall the notation from Section 3, i.e. assume collective variables $\mathbf{z}(\mathbf{x}) = (z_1(\mathbf{x}), \ldots, z_d(\mathbf{x}))$ where $\mathbf{z} : \mathcal{X} \to \mathcal{Z}$, and $\mathcal{Z}$ is $d$-dimensional. Since the log function is continuous, the limit and log can be interchanged and we have

$$\lim_{n\to\infty} \hat{V}(\mathbf{z}, t_n) = -\frac{1}{\beta'} \log \left( \lim_{n\to\infty} \left( \frac{\hat{R}(\mathbf{z}, t_n)}{\hat{N}(\mathbf{z}, t_n) + \epsilon} \right) + \epsilon \right),$$

where, from Algorithm 1:

$$\hat{R}(\mathbf{z}, t_n) = \sum_{i=1}^{n} r(\mathbf{x}_{t_i}) \exp \left( -\sum_{j=1}^{d} \frac{(z_j - z_j(\mathbf{x}_{t_i}))^2}{2\sigma_j^2} \right), \tag{8}$$

$$\hat{N}(\mathbf{z}, t_n) = \sum_{i=1}^{n} \exp\left(-\sum_{j=1}^{d} \frac{(z_j - z_j(\mathbf{x}_{t_i}))^2}{2\sigma_j^2}\right). \tag{9}$$

Since the domain is bounded, we know that for fixed $\mathbf{z}$, both equation 8 and equation 9 have limit at infinity, i.e. $\lim_{n\to\infty} \hat{R}(\mathbf{z}, t_n) = \infty$ and $\lim_{n\to\infty} \hat{N}(\mathbf{z}, t_n) = \infty$. Hence, by Lemma C.1, we have

$$\lim_{n\to\infty} \frac{\hat{R}(\mathbf{z}, t_n)}{\hat{N}(\mathbf{z}, t_n) + \epsilon} = \lim_{n\to\infty} \frac{\hat{R}(\mathbf{z}, t_n)}{\hat{N}(\mathbf{z}, t_n)},$$

provided the limit on the RHS exists. Next, we show that this limit exists by computing it explicitly. The limit can be written

$$\lim_{n\to\infty} \frac{\hat{R}(\mathbf{z}, t_n)}{\hat{N}(\mathbf{z}, t_n)} = \lim_{n\to\infty} \frac{\sum_{i=1}^{n} r(\mathbf{x}_{t_i}) \exp\left(-\sum_{j=1}^{d} \frac{(z_j - z_j(\mathbf{x}_{t_i}))^2}{2\sigma_j^2}\right)}{\sum_{i=1}^{n} \exp\left(-\sum_{j=1}^{d} \frac{(z_j - z_j(\mathbf{x}_{t_i}))^2}{2\sigma_j^2}\right)}.$$

Recall that metadynamics eventually leads to uniform sampling over the domain, independent of the potential. Hence, since $R$ and $\mathbf{z}(\mathbf{x})$ are analytic, by Lemma C.2 we may replace the metadynamics samples $\mathbf{x}_{t_i}$ with samples $\mathbf{u}_i$ from a uniform distribution over $\mathcal{X}$:

$$\lim_{n\to\infty} \frac{\hat{R}(\mathbf{z}, t_n)}{\hat{N}(\mathbf{z}, t_n)} = \lim_{n\to\infty} \frac{\sum_{i=1}^{n} r(\mathbf{u}_i) \exp\left(-\sum_{j=1}^{d} \frac{(z_j - z_j(\mathbf{u}_i))^2}{2\sigma_j^2}\right)}{\sum_{i=1}^{n} \exp\left(-\sum_{j=1}^{d} \frac{(z_j - z_j(\mathbf{u}_i))^2}{2\sigma_j^2}\right)}.$$

In the limit, the ratio of sums with uniform sampling becomes a ratio of integrals:

$$\lim_{n\to\infty} \frac{\hat{R}(\mathbf{z}, t_n)}{\hat{N}(\mathbf{z}, t_n)} = \frac{\int_{\mathcal{X}} r(\mathbf{x}') \exp\left(-\sum_{j=1}^{d} \frac{(z_j - z_j(\mathbf{x}'))^2}{2\sigma_j^2}\right) \mathrm{d}\mathbf{x}'}{\int_{\mathcal{X}} \exp\left(-\sum_{j=1}^{d} \frac{(z_j - z_j(\mathbf{x}'))^2}{2\sigma_j^2}\right) \mathrm{d}\mathbf{x}'}.$$

The limit is therefore a (scaled) convolution of the reward function with a Gaussian in the collective variable space with width vector $\boldsymbol{\sigma}$. Taking the limit $\sigma_j \to 0$ for all $j \in \{1, 2, \ldots, d\}$, the Gaussian convergences to a delta distribution in the collective variable space and we have

$$\lim_{\sigma\to 0} \lim_{n\to\infty} \frac{\hat{R}(\mathbf{z}, t_n)}{\hat{N}(\mathbf{z}, t_n)} = \int_{\mathcal{X}} r(\mathbf{x}') \delta(\mathbf{z} - \mathbf{z}(\mathbf{x}')) \mathrm{d}\mathbf{x}'.$$

*(*) This step also holds for any kernel that becomes distributionally equivalent to a Dirac delta function in the limit that its variance parameter goes to zero. In particular, it also holds for the von Mises distribution that we use in our alanine dipeptide experiment in $\mathbb{T}^2$.*

Finally, we take the limit $\epsilon \to 0$ to obtain

$$\lim_{\epsilon\to 0} \lim_{\sigma\to 0} \lim_{n\to\infty} \hat{V}(\mathbf{z}, t_n) = -\frac{1}{\beta'} \lim_{\epsilon\to 0} \log\left(\int_{\mathcal{X}} r(\mathbf{x}') \delta(\mathbf{z} - \mathbf{z}(\mathbf{x}')) \mathrm{d}\mathbf{x}' + \epsilon\right) \tag{10}$$

$$= -\frac{1}{\beta'} \int_{\mathcal{X}} \log\left(r(\mathbf{x}')\right) \delta(\mathbf{z} - \mathbf{z}(\mathbf{x}')) \mathrm{d}\mathbf{x}' \tag{11}$$

$$= \int_{\mathcal{X}} V(\mathbf{x}') \delta(\mathbf{z} - \mathbf{z}(\mathbf{x}')) \mathrm{d}\mathbf{x}' := V(\mathbf{z}), \tag{12}$$

where we have used the definition $V(\mathbf{x}') = -\frac{1}{\beta'} \log(r(\mathbf{x}'))$ and in the last step we used the definition of the marginal potential energy in the collective variable space. If $\mathbf{z}(\mathbf{x})$ is invertible, then the delta function simplifies to a delta function in the original space and we obtain the original potential instead of the marginal potential. $\qquad\square$

# D   Experiment details

We detail the experimental setup. In Table 2 we summarise the common hyperparameters used in all experiments. In Table 3 we summarise the experiment-specific values of the MetaGFN hyperparameters.

A detailed discussion of the setup follows in sections D.1 - D.4 below, after some general remarks.

Table 2: Common Hyperparameters for All Experiments

| Parameter | Value |
|---|---|
| **General Training** | |
| Training Batches ($B$) | $10^5$ |
| Optimizer | Adam with gradient clipping |
| Learning Rate Schedule | Linear decay from $10^{-3}$ to 0 |
| $\log Z_\theta$ Learning Rate (TB) | Linear decay from $10^{-1}$ to 0 |
| STB Loss $\lambda$ | 0.9 |
| Min Log-Reward Clip | $-10$ |
| Thompson Sampling Heads | 10 |
| Thompson Sampling Bootstrap ($p$) | 0.3 |
| **Replay Buffer** | |
| Capacity | $10^4$ trajectories |
| Storage Reward Threshold | $> 10^{-3}$ |
| Sampling Strategy | Biased (50% from top 30%, 50% from bottom 70%) |
| **MetaGFN** | |
| `freqRB` | 2 |
| `freqMD` | 10 |

Table 3: Experiment-Specific MetaGFN Hyperparameters

| Hyperparameter | Line | Alanine Dipeptide | Grid (2D/3D/4D) |
|---|---|---|---|
| $\Delta t$ (MD time step) | 0.05 | 0.01 | 0.35 |
| $n$ (MD steps) | 2 | 2 | 3 |
| $\beta$ (Inverse temp.) | 1 | 0.4009 | 1 |
| $\gamma$ (Langevin friction) | 2 | 0.1 | 2 |
| $w$ (Hill width) | 0.15 | $10^{-5}$ | 0.10 |
| $\sigma$ or $\kappa$ (KDE bandwidth) | $\sigma = 0.1$ | $\kappa = 10$ | $\sigma = 2$ |
| $\epsilon$ (Reward threshold) | $10^{-3}$ | $10^{-6}$ | $10^{-4}$ |
| Grid Spacing | 0.01 | 0.1 | 0.075/0.3/0.6 |

**General remarks on setting hyperparameters**

In practice, despite having a large number of parameters, the performance of MetaGFN is not highly sensitive to the specific values. The most important hyperparameters are the KDE bandwidth $\sigma$ and the bias height $w$.

*Setting $\sigma$*: Determined by the length-scale of variation of the problem, which is usually known or can be estimated in practice; for the line environment we set $\sigma = 0.1$ (smallest variance peak was 0.1) and for the grid environments $\sigma = 2$ (variance of peaks is 2). These ball-park values immediately gave the expected metadynamics behaviour and did not require any further tuning.

*Setting $w$*: This controls the deposition rate of the Gaussian bias and is linked to freqMD, which determines how often biases are applied. Simulations with w and freqMD such that $w/\text{freqMD} = \text{const}$ show similar behaviour. If $w$ is too low, exploration slows, and if too high, metadynamics becomes unstable. Intermediate $w$ values yield sensible results. We set $w$ by examining an on-policy training run and setting $w$ such that metadynamics explores new modes in a similar timescale to the time it takes the policy to learn to correctly sample the previous mode. As such, it didn't require extensive hyperparameter tuning. An alternative way

to set $w$ would be to schedule it to decrease overtime, as is done in well-tempered metadynamics Barducci et al. (2008). We opted for standard metadynamics to avoid unnecessary complexity in our presentation.

**Remarks on the computational overhead of MetaGFN**

We compare the runtime for training 1000 batches in each environment using an on-policy GFN and MetaGFN in Table 4. Overall, the computational overhead of MetaGFN is practically negligible (typically $< 5\%$) and often statistically insignificant.

We now reason why we expect the cost overhead to be negligible. Let $d$ be the spatial dimension, $B$ be the buffer size and $b$ be the batch size. The KDE + bias update is $O(d)$ computational cost. The cost of adding to the buffer is $O(B)$ and sampling from the buffer is $O(b)$. These costs also scale linearly with the update frequency, which we kept constant throughout all environments (one metad update every 10 backpropagations). In terms of policy error, this frequency was, within statistical significance, just as good as performing a metad update every 2 steps (Figure 8). Importantly, however, an ADAM update step is also $O(d)$ computational cost, since the input size of the first layer scales linearly with $d$. We have found empirically that an ADAM update on a typical multi-layer perceptron (3 hidden layers, 512 hidden units per layer) was more expensive than a metadynamics update, for the same dimension. Further, since ADAM and metad both have cost $O(d)$, this means that we expect MetaGFN to continue to have negligible computational overhead, even in higher dimensions. This is supported by our empirical results in Table 4.

**Compute resources**: Experiments are performed in PyTorch on a desktop with 32Gb of RAM and a 12-core 2.60GHz i7-10750H CPU. For fairness of comparison in runtimes, we use a CPU since our metadynamics update code has not yet been implemented efficiently for GPU architectures.

Table 4: Comparison of runtimes over 5 independent repeats of 1000 batches. The runtime is reported as mean $\pm$ standard deviation in seconds. The final column shows the percentage overhead of MetaGFN or notes if the difference is not statistically significant. All experiments were run using the hyperparameters in Table 3.

| Environment | On-policy GFN (s) | MetaGFN (s) | Overhead / Difference |
|---|---|---|---|
| Line | $46.5 \pm 0.7$ | $45.5 \pm 0.8$ | Not statistically different |
| Alanine | $130.5 \pm 1.4$ | $129.3 \pm 2.7$ | Not statistically different |
| Grid 2D | $73.4 \pm 1.2$ | $76.3 \pm 0.8$ | $+4\%$ |
| Grid 3D | $150.9 \pm 1.3$ | $156.1 \pm 0.6$ | $+3\%$ |
| Grid 4D | $283.8 \pm 1.5$ | $282.6 \pm 1.6$ | Not statistically different |

## D.1 Common experimental parameters

**Training parameters**: For all environments we train for $B = 10^5$ batches use a learning rate with a linear schedule, starting at $10^{-3}$ and finishing at 0. For the TB loss, we train the $\log Z_\theta$ with a higher initial learning rate of $10^{-1}$ (also linearly scheduled). For the STB loss, we use $\lambda = 0.9$, a value that has worked well in the discrete setting (Madan et al., 2022). We use the Adam optimiser with gradient clipping. For all loss functions, we clip the minimum log-reward signal at $-10$. This enables the model to learn despite regions of near-zero reward between the modes of $r(x)$.

**Replay buffer**: We use a replay buffer with capacity for $10^4$ trajectories. Trajectories are stored in the replay buffer only if the terminal state's reward exceeds $10^{-3}$. When drawing a replay buffer batch, trajectories are bias-sampled: 50% randomly drawn from the upper 30% of trajectories with the highest rewards, and the remaining 50% randomly drawn from the lower 70%.

**Noisy exploration**: An additional constant, $\bar{\sigma}$, is added to the standard deviations of the Gaussian distributions of the forward and backward policies. Specifically, the forward policy becomes $\hat{p}_F(s_t, s_{t-1}; \theta) = \sum_{i=1}^{k} w_i \mathcal{N}(\mu_i, (\sigma_i + \bar{\sigma})^2)$, and similarly for the backward policy, where $k$ is the number of Gaussians in the mixture. We schedule the value of $\bar{\sigma}$ so that it decreases during training according to an exponential-flat

schedule:

$$\bar{\sigma} = \begin{cases} \bar{\sigma}_0 \left( e^{-2je/(B/2)} - e^{-2e} \right) & j < B/2 \\ 0 & j \geq B/2, \end{cases} \tag{13}$$

where $j \in (1, \ldots, B)$ is the batch number and $\bar{\sigma}_0 = 2$ is the initial noise, plotted in Figure 7. Note that, for the alanine dipeptide environment, the policy is a mixture of bivariate von Mises distributions and the noise $\bar{\sigma}$ is added to the concentration parameter $\kappa$, where concentration is related to standard deviation through $\sigma = \frac{1}{\kappa^2}$.

**Thompson sampling**: We use 10 heads with the bootstrapping probability parameter set to $p = 0.3$.

**Local Search GFN**: Local search samples trajectories using a forward-backward reconstruction method. First, forward policy samples actions to generate batch size $b$ on-policy trajectories. Then, the backward policy is used to re-sample the last $K$ steps of each trajectory. Finally, the forward policy is used to reconstruct the last $K$ steps, and trajectories in the batch with higher rewards from reconstruction replace the originals. We use $K = 1$ for the line environment, alanine dipeptide environment and two-dimensional grid. We use $K = 2$ and $K = 3$ for the three and four dimensional (hyper)grids, respectively.

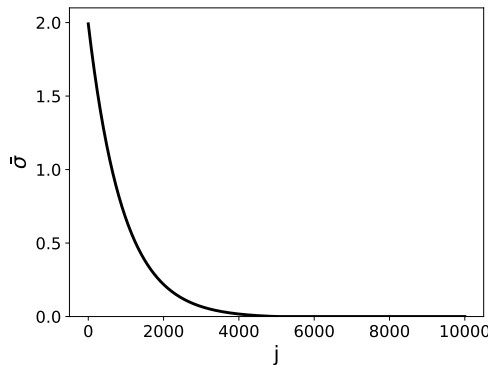

Figure 7: Exponential noise schedule.

**Nested sampling**: We use the Gradient Based Nested Sampling (GBNS) (Lemos et al., 2023) with a publicly available implementation: `https://github.com/Pablo-Lemos/GGNS`. For each environment, GBNS generates samples from the reward distribution. During training, we alternate between two types of backpropagation: one fully on-policy, and the other using backward-sampled trajectories from the generated samples, guided by the current backward policy.

## D.2 Line environment details

**Parameterisation**: The forward and backward policies are a mixture of three Gaussians. We parameterise $\hat{p}_F$, $\hat{p}_B$ and the flow $\hat{f}$ through an MLP with 3 hidden layers, 256 hidden units per layer. We use the GELU activation function and dropout probability 0.2 after each layer. This defines the torso of the MLP. Connecting from this common torso, the MLP has three single-layer, fully-connected heads. The first two heads have output dimension 9 and parameterise the 3 means ($\mu$), standard deviations ($\sigma$) and weights ($w$) of the mixture of Gaussians for the forward and backward policies respectively. The third head has output dimension 1 and parameterises the flow function $\hat{f}$. The mean and standard deviation outputs are passed through a sigmoid function and transformed so that they map to the ranges $\mu \in (-14, 14)$ and $\sigma \in (0.1, 1)$. The mixture weights are normalised with the softmax function. The exception to this parameterisation is the backward transition to the source state which is fixed to be the Dirac delta distribution centred on the source, i.e. $\hat{p}_B(s_0|s_1; \theta) = \delta_{s_0}$. For the TB loss, we treat $\log Z_\theta$ as a separate learnable parameter. We use batch size $b = 64$. It takes approximately 1 hour to train a continuous GFlowNet in this environment with $B = 10^5$ batches.

**MetaGFN**: We use $\Delta t = 0.05$, $n = 2$, $\beta = 1$, $\gamma = 2$, $w = 0.15$, $\sigma = 0.1$, $\epsilon = 10^{-3}$. The domain of Adapted Metadynamics is restricted to $[-5, 23]$ and reflection conditions are imposed at the boundary. The bias and KDE potentials are stored on a uniform grid with grid spacing 0.01. Initial metadynamics samples are drawn from a Gaussian distribution, mean 0 and variance 1, and initial momenta from a Gaussian distribution, mean 0 and variance 0.5.

**Evaluation**: The L1 error between the known reward distribution, $\rho(x) = r(x)/Z$, and the empirical on-policy distribution, denoted $\hat{\rho}(x)$, estimated by sampling $10^4$ on-policy trajectories and computing the empirical distribution over terminal states. Specifically, we compute error $= \frac{1}{2} \int_{-5}^{23} \left| \hat{\rho}(x) - \frac{r(x)}{Z} \right| dx$, where the integral is estimated by a discrete sum with grid spacing 0.01. Note that this error is normalised such that for all valid probability distributions $\hat{\rho}(x)$, we have $0 \leq \text{error} \leq 1$.

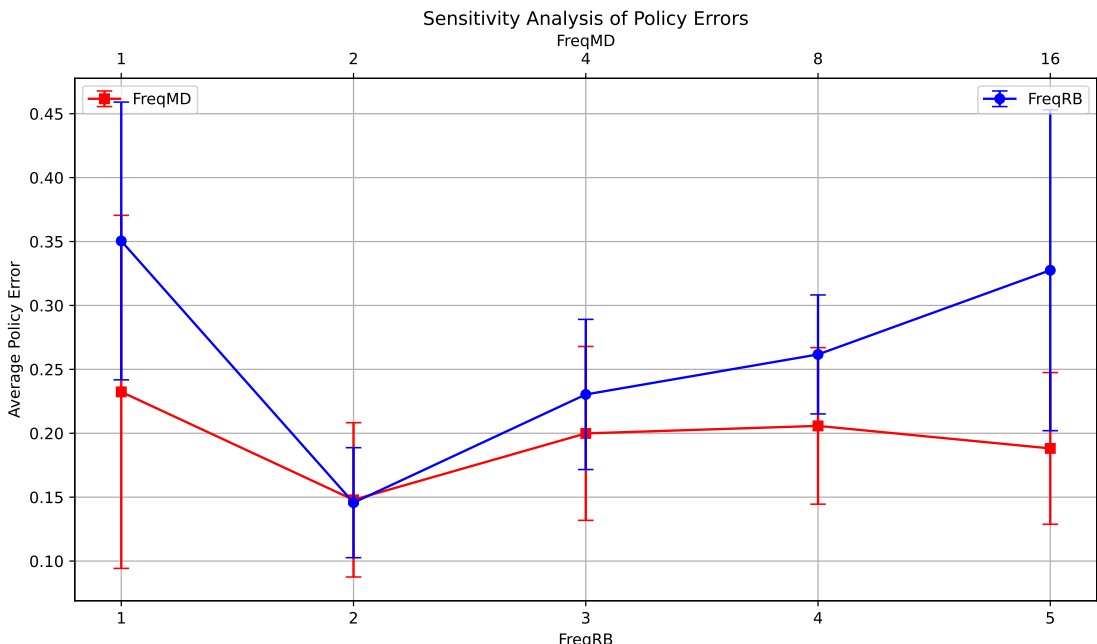

Figure 8: Mean and standard deviation L1 loss after $10^5$ training batches over 10 repeats in the line environment. Blue line: changing `freqRB` while `freqMD = 10`. Red line: changing `freqMD` while `freqRB = 2`.

**On-policy distributions**: Figure 9 shows the forward policy and replay buffer distributions (with bias sampling) after training for $10^5$ iterations with TB loss. MetaGFN is the only method that can uniformly populate the replay buffer and consistently learn all three peaks.

**Adapted Metadynamics**: Figure 10a shows the L1 error between the density implied by the KDE potential and the true reward distribution during a typical training run. Figure 10b shows the resulting $\hat{V}$ and $V_{\text{bias}}$ at the end of the training. By (1), Adapted Metadynamics has fully explored the central peaks. At (2), the third peak is discovered, prompting rapid adjustment of the KDE potential. By $2.5 \times 10^4$ iterations, a steady state is reached and the algorithm is sampling the domain uniformly.

**Sensitivity analysis of hyperparameters**: We compare MetaGFN performance for various `freqRB` and `freqMD` values, showing the mean and standard deviation of the final L1 error after $10^5$ batches (Figure 8). Intuitively, we expect that if `freqRB` is set too large, then MetaGFN focuses predominantly on the on-policy dynamics and may fail to learn to sample distant modes. By contrast, setting `freqRB = 1` means exclusively training on the replay buffer, without any on-policy training trajectories, which we would expect to lead to slow convergence. Therefore, an intermediate value of `freqRB` should be optimal. We found that setting `freqRB = 2` (giving an equal weighting to both on-policy and replay buffer trajectories) showed the lowest loss. By contrast, performance is not too sensitive to `freqMD`. `freqMD` should be set so that Adapted Metadynamics explores the reward landscape in a comparable time it takes to train the GFN to convergence, hence only the order of magnitude of `freqMD` is important. For concreteness, we choose `freqMD = 10` but the model performance is not too sensitive to this parameter.

**Comparing MetaGFN variants**: We consider three MetaGFN variants. The first variant, *always backwards sample*, regenerates the entire trajectory using the current backward policy when pulling from the replay buffer. The second variant, *reuse initial backwards sample*, generates the trajectory when first added to the replay buffer and reuses the entire trajectory if subsequently sampled. The third variant, *with noise*, is to always backwards sample with noisy exploration as per equation equation 13. We plot the L1 policy errors in Figure 11. We observe that *always backward sample* is better than *reuse initial backwards sample* for all loss functions. For DB and TB losses, there is no evidence for any benefit of adding noise, whereas noise improves training for STB loss, performing very similarly to TB loss without noise.

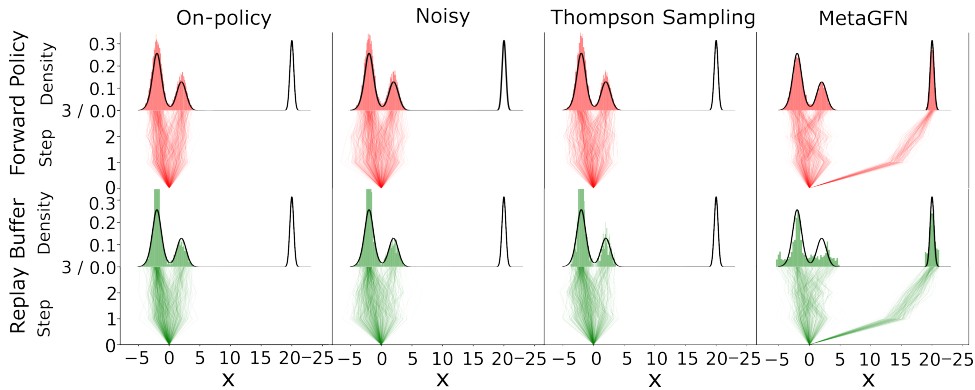

Figure 9: Forward policy and replay buffer distributions after training for $10^5$ iterations with TB loss. MetaGFN is the only method that can consistently learn all three peaks.

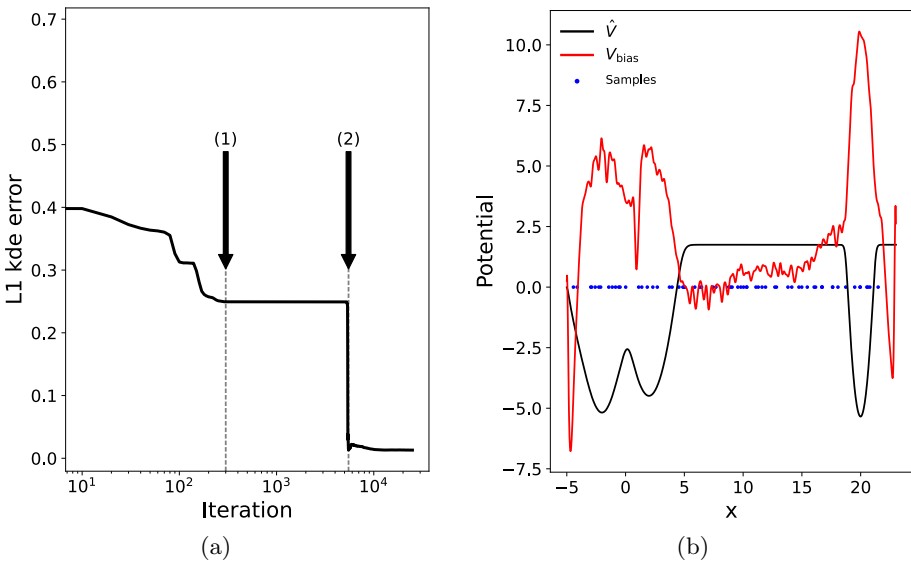

Figure 10: (a) L1 error between $\hat{\rho} = \exp(-\beta\hat{V}(x))/Z$ and the reward distribution, $r(x)/Z$ (b) kde potential, bias potential, and positions of final samples after $2.5 \times 10^4$ training iterations.

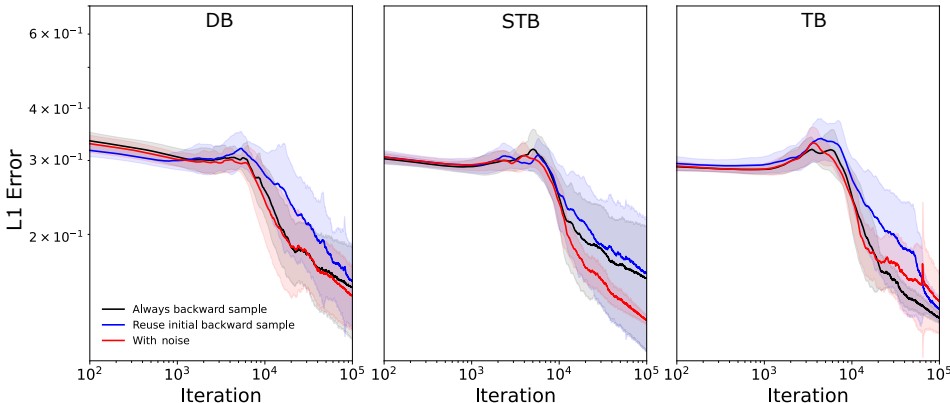

Figure 11: Comparing MetaGFN variants with and without backward sampling in the line environment
.

### D.3 Alanine dipeptide environment details

**Computing the free energy surface**: To obtain a ground-truth free energy surface (FES) in $\phi$-$\psi$ space, we ran a 250ns NPT well-tempered metadynamics MD simulation of alanine dipeptide at temperature 300K ($\beta = 0.4009$), pressure 1bar with the TIP3P explicit water model (Jorgensen et al., 1983). We used the PLUMMED plugin (Bonomi et al., 2019) for OpenMM (Eastman et al., 2017) with the AMBER14 force field (Salomon-Ferrer et al., 2013).

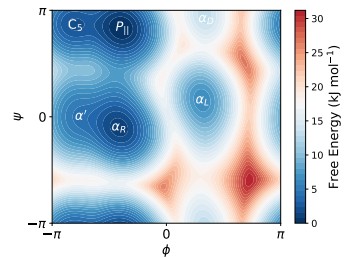

**Parameterisation**: The forward and backward policies are a mixture of three bivariate von Mises distributions. We parameterise $\hat{p}_F$, $\hat{p}_B$ and $\hat{f}$ through three heads of an MLP with 3 hidden layers with 512 hidden units per layer, with GeLU activations and dropout probability 0.2. The first two heads have output dimensions of 15, parameterising the 6 means, 6 concentrations, and 3 weights of the mixture of von Mises policy. The third head has output dimension 1 and parameterises the flow function

Figure 12: The potential KDE $\hat{V}$ learnt using MetaGFN on the alanine dipeptide environment, concentration $\kappa = 10$.

$\hat{f}$. The means are mapped to the range $(-\pi, \pi)$ through $2\arctan(\cdot)$. Concentrations are parameterised in log space and passed through a sigmoid to map to the range $\ln(\kappa) \in (0, 5)$. Mixture weights are normalised with the softmax function. We use batch size $b = 64$. It takes approximately 10 hours to train a continuous GFlowNet in this environment with $B = 10^5$ batches.

**MetaGFN**: We use `freqRB = 2`, `freqMD = 10`, $\Delta t = 0.01$, $n = 2$, $\beta = 0.4009$, $\gamma = 0.1$, $w = 10^{-5}$, $\kappa = 10$, $\epsilon = 10^{-6}$. The bias and KDE potentials are stored on a uniform grid with grid spacing 0.1. Initial samples are drawn from a Gaussian distribution, mean centred $P_{||}$, variance $\sigma^2 = (0.1, 0.1)$, and initial momenta from a Gaussian, mean $\mu = (0, 0)$, variance $\sigma^2 = (0.05, 0.05)$. In Figure 12 we show the resulting learnt KDE potential with these parameters.

**Evaluation**: The L1 error of a histogram of on-policy samples, $\hat{\rho}(\phi, \psi)$, is computed via a two-dimensional generalisation of equation D.2; error $= \frac{1}{2} \int_{-\pi}^{\pi} \int_{-\pi}^{\pi} \left| \hat{\rho}(\phi, \psi) - \frac{r(\phi, \psi)}{Z} \right| \mathrm{d}\phi \mathrm{d}\psi$, estimated by a discrete sum with grid spacing 0.1.

### D.4 Grid environments details

**Parameterisation**: For the two-dimensional grid, we use a mixture of 4 Gaussians. For the three and four dimensional (hyper)grids, we use a mixture of 2 Gaussians (but use longer trajectories, see Section 4). We parameterise $\hat{p}_F$, $\hat{p}_B$ and the flow $\hat{f}$ through an MLP with 3 hidden layers, 512 hidden units per layer, GELU activation function and dropout probability 0.2. The means and standard deviations are mapped to the range $\mu \in (-15, 15)$ and $\sigma \in (0.1, 7)$ through sigmoid functions. Mixture weights are normalised with the softmax function. We use batch size $b = 128$. It takes approximately 10 hours to train a continuous GFlowNet in this environment with $B = 10^5$ batches.

**MetaGFN**: We use `freqRB = 2`, `freqMD = 10`, $\Delta t = 0.35$, $n = 3$, $\beta = 1$, $\gamma = 2$, $w = 0.10$, $\sigma = 2$, $\epsilon = 10^{-4}$. The bias and KDE potentials are stored on a uniform grid with grid spacing 0.075/0.3/0.6 (2, 3, and 4 dimensions respectively). Initial samples are drawn from a Gaussian distribution, mean centred at the origin, isotropic variance $\sigma^2 = 1$. Momenta are initialised to zero.

**Evaluation**: The L1 error between the histogram of the terminal states of $10^4$ on-policy samples and the exact distribution is computed, e.g. for the 2D grid, error $= \frac{1}{2} \int_{-15}^{15} \int_{-15}^{15} \left| \hat{\rho}(x, y) - \frac{r(x, y)}{Z} \right| \mathrm{d}x\mathrm{d}y$, estimated by a discrete sum with grid spacing 0.075.

