# OpenReview forum: "MetaGFN: Exploring Distant Modes with Adapted Metadynamics for Continuous GFlowNets"
_TMLR — Accepted by TMLR_

### Review · Reviewer_f3nt · 2025-05-27

**Summary Of Contributions:**

The authors present MetaGFN, a novel exploration strategy for continuous Generative Flow Networks (GFlowNets, GFNs) based on adaptive Metadynamics. They provide theoretical backing for MetaGFN and empirically demonstrate their proposed approach's effectiveness for exploration in continuous state space environments. The authors showcase that MetaGFN can outperform and accelerate convergence to target reward distributions relative to counterpart exploration strategies. They empirically demonstrate this by considering two synthetic environments, a 1-D "line" environment and continuous hyper-grid environments of varying dimensions, as well as the Alanine dipeptide environment for sampling molecular conformations.

**Audience:**

Yes

**Broader Impact Concerns:**

I have no broader impact concerns.

**Claims And Evidence:**

Yes

**Requested Changes:**

I think this is a very interesting work! I recommend the authors try and make some of the changes/additions pertaining to the items I outlined above in weaknesses. Specifically, adding an additional evaluation metric(s), either one of the metrics I suggested above, or another one from the literature. As well as elaborating on language in some parts of the text.

**Strengths And Weaknesses:**

**Strengths:**

(+) The paper is generally well presented and takes on a novel perspective for exploration when training continuous state space GFlowNets. I believe this is a valuable contribution to the broader area of GFlowNet literature that can be extended and further investigated in future work.
(+) To add, the authors support the claims of their work (MetaGFN) with both theoretical contributions and empirical evidence through a set of several experiments.
(+) From my understanding, I believe this is one of the first works to explore exploration strategies for GFlowNets in continuous state spaces.

**Weaknesses:**

- Some of the language could use clarification in certain parts of the text. For example, the authors state, "However, similar to nested sampling, metadynamics has the drawback of requiring a potential and its gradients, which are not directly available in a black-box GFlowNet setting". It is not clear to me what is meant by this. Do you mean that training via metadynamics exploration requires the gradients of the energy potential? And this is non-trivial to acquire? If this is the case, could the authors expand on why this is the case?
- In a similar vein, after reading the paper, what remains slightly unclear to me is the "meta" aspect of MetaGFN. I understand that at a high-level, the "landscape" of the reward (energy potential) is being adjusted throughout optimization (correct me if my understanding is wrong). However, in the text, it is not entirely clear why this is "meta"? It would be useful to have a sentence or two clearly stating and/or explaining the definition of this term.
- Metrics for performance evaluation are limited -- i.e. only L1 error between the true reward distribution and the on-policy estimated distribution is reported. This makes it difficult to evaluate the robustness of the claims. I would consider either (or both) the (i) KL-divergence between true reward distribution and the on-policy estimated distribution and/or (2) the Jensen-Shannon divergence between true reward distribution and the on-policy estimated distribution. These will give you a non-linear metric for the distance between the estimated and true distributions.
    - In addition, some clarification on the current metric would be beneficial. Is the L1 error computed in the log-probability space or directly in probability space?
- Is my understanding correct that the highest dimensional environment considered in the empirical experiments is 4 dimensions (in the hyper-grid environment)? Have the authors explored the behavior of MetaGFN in even higher dimensions? 10-d, or even 50-d? Or possibly provide some intuition on the scaling behavior of MetaGFN.

---

> ### Author Response · Authors · 2025-06-09
> **Response to reviewer f3nt**
>
> We thank the reviewer for their constructive feedback, and for recognising the novelty, theoretical grounding, and empirical contributions of our work. We appreciate the suggestions on clarity, evaluation metrics, and scalability. We updated the manuscript to address each of the points raised with clarifying text (in red).
>
> > "However, similar to nested sampling, metadynamics has the drawback of requiring a potential and its gradients, which are not directly available in a black-box GFlowNet setting". [...] It is not clear to me what is meant by this.
>
> The metadynamics algorithm requires the gradients of the total energy potential, since the system evolves via Langevin dynamics with force
> \begin{align}
> F(x, t) = - \nabla V_{\text{total}}(x, t) = - \nabla V(x) - \nabla V_{\text{bias}}(x, t),
> \end{align}
> which must be recomputed at each integration step. The biasing force $- \nabla V_{\text{bias}}(x, t)$ drives exploration.
>
> In a black-box setting, we assume that we do not have access to the ‘internals’ of the function parameterisation of r(x), so an analytic expression for $\nabla r(x)$ is not known. Thus, computing $\nabla V(x)$ would require a finite differences approach, requiring $\mathcal{O}(d)$ evaluations of $V(x)$ per step if $V(x)$ is d-dimensional. This is highly costly.
>
> Example: suppose a drug-discovery context where $r(x)$ is the binding strength of a drug conformation x to a target protein. Typically, evaluating $r(x)$ requires running a very long molecular dynamics simulation using e.g. alchemical free-energy methods [1]. In this simulation approach, it is not possible to compute $\nabla V(x)$ directly, instead one would estimate $\nabla V(x)$ by finite differences, running many (expensive) simulations at different perturbed configurations.
>
> This motivates adapted metadynamics (AM), which avoids expensive gradient computations. Note that AM is still valid regardless of whether 1 or 2 hold, but the computational advantages are most dramatic when 1 and 2 are both true.
>
> Manuscript updates:
> - Defined the black box setting, with example, at the end of Section 2.2.
> - Restructured opening paragraphs of Section 3 to reflect this change.
>
> > 'meta' confusion
>
> ‘meta’ stems from the name ‘metadynamics’. The naming motivation being that metadynamics accelerates transitions between the metastable states (minima) of V(x).
>
> Manuscript updates:
> - Footnote at the bottom of page 5 now addresses this point of confusion, clarifying that it is different to meta-learning in ML.
>
> > Request for KL/JS metrics
>
> While KL and JS divergences are standard evaluation metrics, we found L1 divergence to be more practical for evaluating GFlowNets.
>
> GFlowNets tend to learn in "bursts", discovering new modes of the target distribution suddenly. Until a mode is found, its sampled probability is zero, making the KL divergence infinite and rendering convergence plots difficult to interpret.
>
> Further, the sensitivity of KL/JS divergence to low-probability regions requires a vast number of model samples to achieve accurate estimates. Even then, the resulting statistical errors were significantly larger than those for L1, obscuring the results. To compensate for L1's limitations, we provide for alanine dipeptide a qualitative breakdown of which modes are sampled.
>
> > L1 clarification
>
> This is computed directly in the probability space. In Appendix D, we give details of this calculation in each environment ('evaluation' paragraphs, D.1-D.4).
>
> > Higher-dimensional experiments
>
> Dimensionality scaling is the biggest weakness of metadynamics; 50-d would be intractable with the current approach. We highlighted this point several times:
>
> - Final paragraph of section 2.4: “Since biases are typically specified on a numerical grid, this gives rise to a memory cost that grows exponentially with dimension. Therefore, biases are typically applied along low-dimensional coordinates known as collective variables (CVs).”
> - End of page 9: “For MetaGFN, this is also an expected outcome of the curse of dimensionality of the replay buffer grid; in high-dimensional spaces metadynamics samples encounter high-reward modes less frequently, leading to a replay buffer with less sample diversity.”
> - Limitations: “For metadynamics to be an effective sampler, the CVs must be low-dimensional and bounded.”
>
> Scaling exploration to larger dimension systems remains an open challenge in molecular dynamics. However, in that context, even if the system has high intrinsic dimension, remarkably that does not typically preclude finding a small dimensional subspace in which sampling can be accelerated [2]. This is why we believe our contribution is impactful, even with this limitation.
>
> Manuscript updates:
> - We expanded the discussion in the limitations (Sec. 5) and we have added a citation to reconnaissance metadynamics, a method that is known to have better dimension scaling.
>
> [1] arxiv.org/abs/2008.03067
> [2] arxiv.org/abs/2202.04164

---

### Review · Reviewer_rMtr · 2025-06-05

**Summary Of Contributions:**

This paper integrates metadynamics into the framework of GFlowNets to enhance the exploration of distant modes in continuous settings. It is shown that the proposed method outperforms existing baseline exploration techniques in toy continuous experiments.

**Audience:**

Yes

**Claims And Evidence:**

Yes

**Requested Changes:**

It is recommended to include additional experiments on high-dimensional, real-world scenarios.

**Strengths And Weaknesses:**

Strengths:
This work introduces a theoretically grounded technique to enhance the exploration capabilities of continuous GFlowNets, supported by experimental validation.

Weaknesses:
The experiments are limited to low-dimensional toy problems, providing insufficient evidence for the method’s effectiveness in more challenging, real-world scenarios.

---

> ### Author Response · Authors · 2025-06-23
> **Response to reviewer rMtr**
>
> We thank the reviewer for their feedback, and recognising that our proposed technique is theoretically grounded and supported experimentally. The identified weakness was that the method was limited to lower-dimensional settings, that these are not real-world scenarios, and that we should perform additional experiments in these settings. We believe that this criticism is only partially valid, and in practice, it would not be possible to easily perform the experiments that the reviewer might be suggesting, as we now explain below.
>
> Our argument can be summarised as:
> 1. Metadynamics is inherently limited to exploration in a low-dimensional subspace, hence so is MetaGFN.
> 2. However, so long as the low-dimensional exploration manifold can be identified, this does not preclude using MetaGFN in high-dimensional, real-world settings.
> 3. Positioning: we adapt an existing exploration strategy from molecular dynamics, and demonstrate that it is superior to alternatives for training GFNs, *within the limitations of the exploration method*. We do not claim to have a definitive SOTA in all application settings.
>
> **Expanding on point 1**:
>
> The classical metadynamics algorithm requires storing a bias potential on a grid. Inevitably, the memory cost of storing this grid grows exponentially with dimension. The grid therefore has to be very low resolution in high-dimensional spaces. The effectiveness of MetaGFN is therefore expected to degrade with dimension, as the corresponding KDE estimate becomes a poor approximation for the underlying potential. This issue could be mitigated by learning a more efficient representation than a grid, for instance by training an autoencoder to learn the mapping to the collective variable space, and jointly learning a parametric function on the collective variables to predict the bias potential by minimising a least squares loss. However, this regards collective variable learning strategies, not exploration, and lies outside the scope of our work (see point 2).
>
> **Expanding on point 2**:
>
> Many real-world problems, for instance in molecular conformation sampling, involve intrinsically high-dimensional systems but can nonetheless be effectively sampled by exploring a low-dimensional submanifold. The alanine dipeptide system we considered is such an example, having 22 atoms and thus being a 66-dimensional system, but for which there is a low-dimensional exploration manifold described by only 2 collective variables (the two backbone angles).
>
> There is a rich literature on how to learn collective variables for systems of interest [1,2]. In molecular systems, they correspond to the slow degrees of freedom. We consider the task of learning the collective variables out of scope for our work. Our starting point is that they have already been identified.
>
> [1] Machine learning for collective variable discovery and enhanced sampling in biomolecular simulation, Sidky H., Chen W., Ferguson A., Molecular Physics (2020)
>
> [2] Collective variable-based enhanced sampling and machine learning, Chen M., Topical Review - Computational Methods, The European Physics Journal B (2021)
>
> **Discussion in the manuscript**:
>
> We already mentioned aspects of these points in the original manuscript: At the beginning of the final paragraph of section 2.4: “Since biases are typically specified on a numerical grid, this gives rise to a memory cost that grows exponentially with dimension. Therefore, biases are typically applied along low-dimensional coordinates known as collective variables (CVs).”. At the end of page 9: “For MetaGFN, this is also an expected outcome of the curse of dimensionality of the replay buffer grid; in high-dimensional spaces metadynamics samples encounter high-reward modes less frequently, leading to a replay buffer with less sample diversity.” As well as at the beginning of the limitations section: “For metadynamics to be an effective sampler, the CVs must be low-dimensional and bounded.”. We have now also expanded our discussion in the limitations section to improve the positioning of the work. If you feel that more changes are needed to improve positioning, please let us know.

---

### Review · Reviewer_6BLR · 2025-06-23

**Summary Of Contributions:**

This paper proposes MetaGFN, a novel method that adapts metadynamics-inspired exploration strategies to continuous Generative Flow Networks (GFlowNets). The key idea is to bias sampling trajectories using a KDE-based repulsive potential in a learned or user-supplied collective variable (CV) space, enabling better discovery of distant high-reward modes. The method is theoretically grounded with a consistency result (Theorem 3.1) and is empirically validated on synthetic multimodal benchmarks.

**Audience:**

Yes

**Broader Impact Concerns:**

I do not foresee major ethical or societal concerns arising from this work. On the contrary, the method may have positive implications in scientific discovery tasks, like molecular design, rare-event simulation, or multimodal inference.

**Claims And Evidence:**

Yes

**Requested Changes:**

1. It would be good to consider a table to summarize all hyperparameters used in the experiments (e.g., bias height, width, stride, and replay buffer size).
2. While the paper states that the computational overhead of Adapted Metadynamics is "<5%" in all experiments, this claim is not quantitatively backed. (1) Which parts of the algorithm (KDE computation, bias update, buffer maintenance) contribute most to runtime and memory cost? (2) Whether the overhead remains negligible in higher dimensions or with more frequent bias updates? (3) How does the overhead compare between MetaGFN and baseline GFlowNet across tasks?
3. Check out some typos like on Page 6 "access to not only the potential potential", Page 8 "a a small biomolecule of 23 atoms".

**Strengths And Weaknesses:**

**Strengths**
- It introduces a principled way to incorporate metadynamics–style bias into continuous GFlowNets, bridging sampling methods from molecular dynamics with generative modeling.
- It provides a rigorous consistency proof (Theorem 3.1) for the KDE‐based estimator of the repulsive potential.

**Weaknesses**
- All experiments are in low dimensions; it remains unclear how the grid‐based KDE and replay buffer will scale to moderate or high dimensions.
- Critical parameters (KDE bandwidth σ, bias height w, update stride) lack systematic sensitivity analysis or adaptive tuning strategies.
- Overhead is only briefly mentioned (< 5 % in low‐D); a detailed breakdown of computational and memory costs versus dimension would aid reproducibility and deployment.

---

> ### Author Response · Authors · 2025-07-05
> **Response to reviewer 6BLR**
>
> We thank the reviewer for their constructive feedback highlighting both the theoretical strengths of our work and concerns regarding scalability, overhead analysis, and hyperparameter reporting. We now address these points, providing quantitative details and clarifications.
>
> > All experiments are in low dimensions; it remains unclear how the grid‐based KDE and replay buffer will scale to moderate or high dimensions.
>
> We acknowledge our work's limitation to low-dimensional exploration, but we emphasise that this only pertains to the exploration manifold, not necessarily the problem's overall dimensionality. High-dimensional problems are therefore tractable, provided an efficient, low-dimensional manifold can be found. Regarding scaling, while a grid-based KDE for the bias potential has exponential memory scaling with dimension, this issue could be mitigated by defining the bias potential parametrically instead. We discuss these aspects in greater detail in our response to Reviewer rMtr.
>
> The required scaling for the replay buffer is less clear, although note that we found a fixed buffer size of 10,000 trajectories to be adequate across all our experiments (1D-4D).
>
> > Critical parameters (KDE bandwidth σ, bias height w, update stride) lack systematic sensitivity analysis or adaptive tuning strategies.
>
> It is a valid point that we do not have an automated strategy to set several of the hyperparamers. However, we found in practice that the result of the method was not highly sensitive to the specific choices of w and σ; only the right order of magnitude is sufficient for reasonable results.
>
> Setting σ: We mentioned in Section 3 that σ is determined by the length-scale of variation of the problem, which is usually known or can be estimated in practice; for the line environment we set σ = 0.1 (smallest variance peak was 0.1) and for the grid environments σ = 2 (variance of peaks is 2). We did not require tuning - these ball-park values immediately gave the expected metadynamics behaviour.
>
> Setting w: This parameter controls the deposition rate of the Gaussian bias and is linked to freqMD, which determines how often biases are applied. Simulations with w and freqMD such that w/freqMD = const show similar behaviour. If w is too low, exploration slows, and if too high, metadynamics becomes unstable. Intermediate w values yield sensible results. We set w by examining an on-policy training run and setting w such that metadynamics explores new modes in a similar timescale to the time it takes the policy to learn to correctly sample the previous mode. As such, it didn't require any extensive hyperparameter tuning. An alternative way to set w in a principled way would be to schedule it to decrease overtime, as is done in well-tempered metadynamics [1]. We opted for standard metadynamics to avoid unnecessary complexity in our presentation.
>
> [1] Barducci et al, 2008
>
> **We have now added this discussion to the beginning of Appendix D (Experimental details).**
>
> > It would be good to consider a table to summarize all hyperparameters used in the experiments (e.g., bias height, width, stride, and replay buffer size).
>
> All of these hyperparameters were specified in Appendix D but we agree that the information is a bit diffuse. **We have now added two tables at the beginning of Appendix D: Table 2 details the common hyperparameters across all experiments and Table 3 details the different MetaGFN hyperparameters used in different experiments.**
>
> >(1) Which parts of the algorithm (KDE computation, bias update, buffer maintenance) contribute most to runtime and memory cost?
> > (2) Whether the overhead remains negligible in higher dimensions or with more frequent bias updates?
> > (3) How does the overhead compare between MetaGFN and baseline GFlowNet across tasks?
>
> Let d be the spatial dimension, B be the buffer size and b be the batch size.
> Storing the bias grid is cost O(2^d) in memory.
> The KDE + bias update is O(d) computational cost.
> The cost of adding to the buffer is O(B) and sampling from the buffer is O(b).
> The cost of the aforementioned updates of course scales linearly with the update frequency. However, we kept that constant throughout all of our tasks (one metad update every 10 backpropagations), and indeed this frequency was, within statistical significance, just as good as performing a metad update every 2 steps (Figure 8; Appendix D).
>
> An ADAM update step is O(d) computational cost (size of input in first layer grows linearly with dimension). We have found empirically that this the most expensive step. Since ADAM and metad both have cost O(d), this means that we expect MetaGFN to continue to have negligible computational overhead, even in higher dimensions (although the memory overhead is severe).
>
> **We have added this discussion as well as a timing benchmark comparing the cost of training in each environment of on-policy GFNs vs MetaGFN in Table 4, Appendix D.**
>
> > Check out some typos.
>
> Thank you, fixed now.

---

### Decision · Action_Editor_7snY · 2025-09-09

**Recommendation:** Accept as is

**Additional Comments:**

One of the reviewers highlighted how this paper provides all the necessary information to reproduce the paper and also provides the code to do so. The appendix discusses hyperparameter choices and experimental setup with enough details. So I recommend a reproducibility certificate for this paper.

**Audience:**

Yes

**Audience Explanation:**

The paper would interest the GFlowNets community, and there will be enough readers to read it.

**Claims And Evidence:**

Yes

**Claims Explanation:**

This paper introduces a method to improve exploration in continuous GFlowNets which is theoretically grounded. Even though the experiments are with low dimensions, authors clearly acknowledge this limitation. All the reviewers agree to accept this paper.